# DUSP10 constrains innate IL-33-mediated cytokine production in ST2[hi] memory-type pathogenic Th2 cells

Takeshi Yamamoto[1,2], Yusuke Endo[1,3], Atsushi Onodera[1], Kiyoshi Hirahara[1], Hikari K. Asou[1],
Takahiro Nakajima[1,3], Toshio Kanno[1,3], Yasuo Ouchi[4], Satoshi Uematsu[4], Hiroshi Nishimasu[5], Osamu Nureki [5],
Damon J. Tumes [6], Naoki Shimojo[2] & Toshinori Nakayama[1]

ST2[hi] memory-type Th2 cells are identified as a pathogenic subpopulation in eosinophilic airway inflammation. These ST2[hi] pathogenic Th2 cells produce large amount of IL-5 upon T cell receptor stimulation, but not in response to IL-33 treatment. By contrast, IL-33 alone induces cytokine production in ST2[+] group 2 innate lymphoid cells (ILC2). Here we show that a MAPK phosphatase *Dusp10* is a key negative regulator of IL-33-induced cytokine production in Th2 cells. In this regard, *Dusp10* is expressed by ST2[hi] pathogenic Th2 cells but not by ILC2, and *Dusp10* expression inhibits IL-33-induced cytokine production. Mechanistically, this inhibition is mediated by DUSP10-mediated dephosphorylation and inactivation of p38 MAPK, resulting in reduced GATA3 activity. The deletion of *Dusp10* renders ST2[hi] Th2 cells capable of producing IL-5 by IL-33 stimulation. Our data thus suggest that DUSP10 restricts IL-33-induced cytokine production in ST2[hi] pathogenic Th2 cells by controlling p38-GATA3 activity.

[1] Department of Immunology, Graduate School of Medicine, Chiba University, 1-8-1 Inohana Chuo-ku, Chiba 260-8670, Japan. [2] Department of Pediatrics, Graduate School of Medicine, Chiba University, 1-8-1 Inohana Chuo-ku, Chiba 260-8670, Japan. [3] Laboratory of Medical Omics Research, KAZUSA DNA Research Institute, 2-6-7 Kazusa Kamatari, Kisarazu, Chiba 292-0818, Japan. [4] Department of Mucosal Immunology, Graduate School of Medicine, Chiba University, 1-8-1 Inohana Chuo-ku, Chiba 260-8670, Japan. [5] Department of Biological Sciences, Graduate School of Science, The University of Tokyo, 2-11-16 Yayoi, Bunkyo, Tokyo 113-0032, Japan. [6] Centre for Cancer Biology, University of South Australia and SA Pathology, cnr North Terrace & Morphett St bridge, GPO Box 2471, Adelaide, SA 5001, Australia. These authors contributed equally: Takeshi Yamamoto, Yusuke Endo. Correspondence and requests for materials should be addressed to T.N. (email: tnakayama@faculty.chiba-u.jp)

The prevalence of allergic diseases such as atopic dermatitis, asthma, and allergic rhinosinusitis has been increasing worldwide, and is a significant public problem in most developed countries[1]. Asthma is one of the most common chronic inflammatory disorders, which is categorized as a lower airway respiratory disease with recurrent wheezing and airway obstruction[2,3]. Allergic asthma is mainly driven by T helper 2 (Th2)-type inflammation including IL-4, IL-5, and IL-13 production, and is characterized by the presence of elevated numbers of eosinophils in the lungs[4–6]. Tissue-derived cytokines including IL-25, IL-33, and thymic stromal lymphopoietin (TSLP) have also been implicated in Th2-associated disease exacerbation by amplifying Th2 cytokine-mediated immune responses[7–10]. ILC2 rapidly respond to tissue-derived cytokines, and produce large amounts of IL-5 and IL-13[11–13]. Furthermore, several recent findings suggested that ILC2 play an important role in eosinophilic airway inflammation in mice that lack the ability to mount adaptive immune responses[14,15]. Th2 cells and ILC2 may contribute to distinct types of inflammation such as allergen-specific and non-specific allergic inflammation, respectively[16]. Th2 cells and ILC2s are the major source of Th2 cytokines in allergic asthma, and these cells may also collaborate in allergen-driven innate and adaptive type2-lung inflammation[17,18]. Very recently, a redundant role for ILC2 in humans has also been proposed[11].

Several subpopulations of memory CD4[+] T cells are implicated in the pathogenesis of chronic inflammatory diseases, including asthma[19,20]. We have identified ST2[+] allergen-specific memory-type pathogenic Th2 (Tpath2) cells in allergic eosinophilic airway inflammation[19,21,22]. We demonstrated that the Tpath2 cells express high levels of ST2, and that IL-33-mediated activation of the p38 MAPK pathway augments the pathogenicity of Tpath2 cells in allergic airway inflammation in both mice and humans[23]. Interestingly, IL-33 induced chromatin remodeling at the *Il5* gene locus in memory Th2 cells is independent of TCR signaling. However, IL-33 alone does not induce IL-5 and IL-13 expression and production in Tpath2 cells. ILC2s also express high levels of ST2, but in contrast to Tpath2 cells, ILC2s rapidly produce large amounts of IL-5 and IL-13 in response to IL-33[12,24]. Tpath2 cells and ILC2s share several cardinal features, including cell surface molecule expression (ST2, IL-7Rα, and ICOS), effector function in terms of IL-5 and IL-13 cytokine production, and the expression of key signaling molecules (e.g., p38 MAPK) and transcriptional factors (e.g., GATA3) relevant to their differentiation and function[21]. Despite these similarities, fundamental differences in the regulation of cytokine production exist with selective production of IL-4 by Tpath2 cells, and IL-33 able to directly induce cytokine production from ILC2s. The molecular mechanisms that control these functional differences between adaptive and innate lymphocytes remain unclear.

GATA3, known as a master transcription factor for Th2 cell differentiation, directly transactivates the *Il5* and *Il13* genes[25,26]. The activation and nuclear translocation of GATA3 are dependent on its phosphorylation on serine residues induced by p38 MAPK[26,27]. Both p38 and GATA3 are known to be required for the production of IL-5 and IL-13 by ILC2s. IL-33 induces p38 activation and thereby phosphorylation of GATA3 in ILC2s, and the phosphorylated-GATA3 binds to the *Il5* and *Il13* promoters[28]. Interestingly, some studies suggested that Th2 cells produced IL-13 in an antigen-independent manner[29]. Multiple-rounds of priming in Th2 cells can result in Gata3-dependent IL-13 production in response to IL-33 together with a STAT5 activator[30], suggesting that these transcription factors may induce innate functions of Th2 cells under specific conditions. However, the underlying molecular mechanism by which antigen-specific Th2 cell constrains innate immune function remains unknown.

Activation of MAPK signaling by a variety of agents induces inflammatory responses, including production of proin-flammatory cytokines such as TNF, IL-1β, IL-6, IL-12, and IFNγ[31]. These rapid inflammatory responses protect the host during the initial phase of an infection, but dysregulated cytokine production may also lead to detrimental effects, as is the case during sepsis and septic shock[32]. Activation of the proin-flammatory signaling cascade also triggers negative-feedback mechanisms, which can constrain and terminate the rapid inflammatory response. It has been recognized that MAPK are negatively regulated by dual specificity protein phosphatases (DUSPs, also known as MAPK phosphatase, MKP)[33]. The DUSP family in mammalian cells consists of at least 30 typical and atypical DUSPs that show different patterns of tissue expression, transcriptional control, substrate specificity, and subcellular localization[33].

In the current study, we identify a MAPK phosphatase DUSP10, as a key negative regulator of IL-33-induced IL-5 and IL-13 production in ST2[+] Tpath2 cells. *Dusp10* is highly expressed in the majority of Tpath2 cells, whereas its expression is limited in ILC2s. DUSP10 negatively controls IL-33–ST2 signaling through dephosphorylation of p38 MAPK in Tpath2 cells. Thus, DUSP10 distinguishes the innate IL-33 inducible functions of ILC2 and the adaptive functions of Th2 cells, and modulates the ability of Tpath2 cells to directly produce IL-5 and IL-13 in response to IL-33.

## Results

**Tpath2 cells show limited IL-33-induced production of IL-5.** In the current report, we use the name "Tpath2 cells" to define ST2[hi] memory-type (CD44[hi]CD62L[lo]) Th2 cells that are pathogenic in terms of the induction of IL-5-mediated eosinophilic inflammation[19,21]. To investigate the molecular mechanisms that control cytokine-induced cytokine production in Tpath2 cells and ILC2s, we first assessed the cytokine production profiles of these cells in response to IL-33 and IL-7 plus IL-33. We prepared ST2[hi] memory-type Th2 (Tpath2) cells and ILC2s as demonstrated in Supplementary Fig. 1a–c[23,34]. These cells from the spleen were cultured with IL-7, IL-25 and IL-33 (Supplementary Fig. 1a). As shown in Fig. 1a, stimulation with IL-7 plus IL-33 for 6 h selectively induced the production of IL-5 and IL-13 by ILC2s but induced only very little production of these cytokines by Tpath2 cells (Fig. 1a, top panel, $n = 4$ for each group, $*p < 0.05$, $**p < 0.01$). In contrast, Tpath2 cells produced large amounts of IL-5 and IL-13 following stimulation with PMA and Ionomycin (Fig. 1a, bottom right, $n = 4$ for each group, $*p < 0.05$, $**p < 0.01$). Similar results for IL-5 and IL-13 were obtained by qPCR and ELISA analyses (Fig. 1b, c). *Il4* mRNA expression and IL-4 protein expression were not detected in ILC2s upon IL-7 plus IL-33 stimulation in our experimental system.

We next analyzed splenic CD44[hi] memory phenotype CD4[+] (MPCD4) T cells from normal untreated BALB/c mice maintained under SPF conditions (Supplementary Fig. 1d). The proportions of IL-5 and IL-13 producers within the ST2 expressing population of MPCD4 T cells (ST2[hi] MPCD4 T cells) were ~10–15%, suggesting that these population may also contain non-Th2 ST2[hi] cells such as Th1, Th17 and Treg cells. The ST2[hi] MPCD4 T cells did not produce IL-5 and IL-13 in response to IL-7 plus IL-33, whereas ILC2s did (Fig. 1d, $n = 4$ for each group, $*p < 0.05$).

To explore the mechanisms controlling the distinct outcomes of IL-33–sensing between Tpath2 cells and ILC2s, we first assessed the expression of cytokine receptors between these cells. Indeed, cell surface expression of ST2 on Tpath2 cells was higher

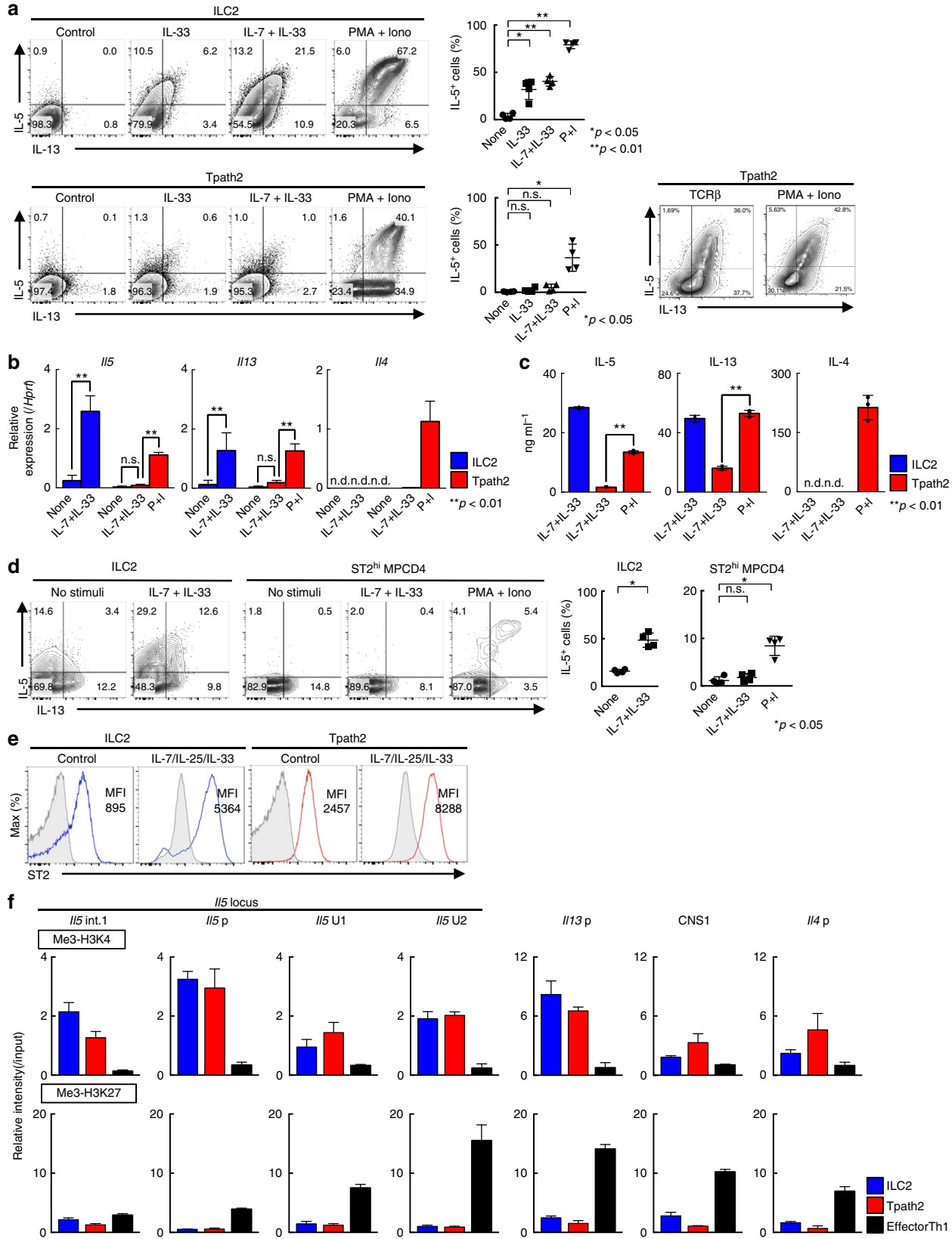

than that on ILC2s with or without IL-7/IL-25/IL-33 cultivation (Fig. 1e). Similarly, gene expression of *Il1rl1* (*ST2*) and *Il7ra* was almost equivalent between Tpath2 cells and ILC2s (Supplementary Fig. 1e). Chromatin immunoprecipitation (ChIP) assays (Supplementary Fig. 1f shows position of genomic features analyzed) revealed that at the *Il5* locus Tpath2 cells and ILC2s showed increased levels of histone modifications associated with active transcription (H3-K4 trimethylation) and lower modifications associated with gene repression (H3-K27 trimethylation) as compared to effector Th1 cells (Fig. 1f). Similarly, the levels of

**Fig. 1** Tpath2 cells show limited IL-33-induced production of IL-5. To compare Tpath2 cells and ILC2s, both cell subsets were prepared from the spleen of memory Th2 mice (see Supplementary Fig. 1a and Methods section). Cells were cultured with IL-7 in complete medium for 1 day after cell sorting, then stimulated with indicated stimuli for 6 h. **a** Intracellular-staining profiles of IL-5 and IL-13 in Tpath2 cells and ILC2s are shown. The cells were stimulated with indicated stimuli for 6 h. The percentages of IL-5 positive ILC2s and Tpath2 cells are shown (mean ± SD; $n = 4$ for each group). **b** Quantitative RT-PCR analysis of the indicated cytokines in Tpath2 cells and ILC2s. The cells were stimulated with indicated stimuli for 4 h. Relative expression (normalized to *Hprt*) is shown with standard deviation. **c** ELISA analysis of the indicated cytokines in the culture supernatant from $5 \times 10^4$ cells of Tpath2 cells and ILC2s, respectively. The cells were stimulated with indicated stimuli for 12 h. **d** Intracellular-staining profiles of IL-5 and IL-13 in ST2$^{hi}$ MPCD4 T cells and ILC2s purified from normal BALB/c mice are shown and stimulated with the indicated stimuli for 6 h. The percentages of IL-5 positive ILC2s and ST2$^{hi}$ MPCD4 T cells are shown (mean ± SD; $n = 4$ for each group). **e** Expression profiles of ST2 on Tpath2 cells and ILC2s before (immediately after sorting) and after cultivation with IL-7, IL-25 and IL-33 are shown. The number in the histogram represents mean fluorescence intensity (MFI). Gray filled histogram shows isotype control staining. **f** ChIP assays were performed with anti-trimethyl histone H3-K4 and anti-trimethyl histone H3K27 at the Th2 cytokine gene loci from Tpath2 cells, ILC2s, and effector Th1 cells. The relative intensities (relative to input DNA) of these modifications were determined by quantitative RT-PCR analysis. More than five for **a** to **e**, and three for **f** independent experiments were performed and showed similar results (**$p < 0.01$; n.s. not significant, n.d. not detected; Mann–Whitney $U$ test; **a**–**c**). Three technical replicates were performed with quantitative RT-PCR (**b**, **f**) and ELISA (**c**)

H3K4 trimethylation at the *Il13*p or other Th2 cytokine loci including CNS1 and *Il4*p were higher in Tpath2 cells and ILC2s than those of effector Th1 cells (Fig. 1f and Supplementary Fig. 1g shows position of genomic features analyzed). These results indicate that chromatin structure at the *Il5* gene locus was maintained in a permissive status both in Tpath2 cells and ILC2s. Collectively, IL-33-induced IL-5 and IL-13 production by Tpath2 cells was very limited as compared to ILC2s, even though Tpath2 cells had high-level expression of ST2 and permissive chromatin status at the *Il5* and *Il13* loci. These data confirm that Th2 cytokine production by Tpath2 cells requires TCR stimulation with antigens and does not readily occur in response to IL-33 exposure.

**Preferential expression of *Dusp10* in Tpath2 cells.** We performed RNA sequencing analysis using Tpath2 cells and ILC2s, and selected transcripts that were expressed at appreciable levels (Fragments per kilobase of exon per million fragments mapped: FPKM > 10) in Tpath2 cells, and then, the expression levels were compared between Tpath2 cells and ILC2s (Supplementary Fig. 2a). Lineage markers such as CD antigens and some noteworthy genes showed higher FPKM values selectively in Tpath2 cells as compared with ILC2s (genes above eight-fold change ratio shown in Supplementary Fig. 2a). In addition, Tpath2 cells and ILC2s share several common Th2 response-associated genes including cell surface markers, cytokines, and transcription factors that have been previously described (*Icos, Crlf2, Il9r, Il1rl1*, and *Gata3*) (Fig. 2a)[34–36]. Almost all lineage markers detected in Fig. 2a were shared with effector Th2 cells (Supplementary Fig. 2b). An analysis of gene ontology by DAVID showed a significant enrichment in immunological processes (Fig. 2b). Among them, different expression of the genes involved in the pathway of the "MAP kinase tyrosine/serine/threonine phosphatase activity" appeared to be frequently observed in Tpath2 cells but not ILC2s in the molecular function (MF) category (Fig. 2b, MF). Since both Tpath2 cells and ILC2s show activation of the p38 MAKP pathway after IL-33 stimulation[34–36], we focused on the expression of the MAP kinase phosphatases: Dual-specificity phosphatase (DUSP). Dozens of DUSPs family member have been identified in the last several years, and their expression profiles and substrate specificities depend on cell types[37]. Therefore, we compared the expression profiles of DUSPs between Tpath2 cells and ILC2s (Fig. 2c). We detected higher expression of *Dusp10* in Tpath2 cells as compared to ILC2s (Fig. 2c, Supplementary Fig. 2a–c). In addition, *Dusp2* expression was slightly higher in Tpath2 cells. However, the expression levels of other *Dusp* genes were largely comparable between Tpath2 cells and ILC2s. We also checked the gene expression profiles of MAPK and upstream kinases involved in MAPK signaling (*Map2k, Map3k*, and

*Map4k*) (Supplementary Fig. 2d). Genes encoding members of MAPK were equivalently expressed in Tpath2 cells and ILC2s. These results indicate that *Dusp10*, known to be a negative regulator for p38 signaling was selectively and substantially expressed in Tpath2 cells but not in ILC2s.

To quantitatively confirm these findings at the transcriptional level, we next analyzed mRNA expression of *Dusp* family members including *Dusp10*, other phosphatases of MAPK and their substrates in freshly prepared Tpath2 cells and ILC2s. We found that the expression of *Dusp10* in Tpath2 cells was much higher than ILC2s, as was detected by RNA-sequencing analysis (Fig. 2d). Similar results were obtained from assessment of expression of *Dusp10* in ST2$^{hi}$ MPCD4 T cells (Supplementary Fig. 2e). We also examined the expression level of *Dusp10* in effector Th1, Th2, Th17, ILC1s, and ILC3s, and detected lower levels of *Dusp10* in effector Th1, Th2 and Th17 cells as compared to Tpath2 cells. Interestingly, the expression level of *Dusp10* was considerably higher in ILC1s and ILC3s compared to ILC2s (Supplementary Fig. 2f). The expression of other phosphatases, other *Dusp* members, and their substrates was almost equivalent between Tpath2 cells and ILC2s (Fig. 2d). *Dusp2* showed higher expression in Tpath2 cells as compared to ILC2s in Fig. 2c, but the expression level assessed by qPCR analysis was very low. Similarly, IL-33–ST2 signaling-associated kinases including *Myd88, Irak1, Irak4, Traf6*, and *Tab1* were expressed equally in Tpath2 cells and ILC2s (Supplementary Fig. 2g). These results led us to postulate that DUSP10 negatively regulates IL-33–ST2 signaling and IL-33-induced cytokine production in Tpath2 cells.

**Deletion of *Dusp10* increases IL-5 production in Tpath2 cells.** We next assessed the phosphorylation levels of p38 in ILC2s and Tpath2 cells stimulated with IL-7 plus IL-33 or PMA with Ionomycin. Phosphorylation of p38 was clearly induced by IL-7 plus IL-33 in ILC2s (Fig. 3a, b, $n = 3$ for each group, *$p < 0.05$), while the level of IL-7/33-induced phosphorylation of p38 was marginal in Tpath2 cells. These results indicate that IL-33-induced p38 phosphorylation was impaired in Tpath2 cells as compared to ILC2s. Next, to assess the role of *Dusp10* in IL-33-induced IL-5 production in Tpath2 cells, we established a CRISPR/Cas9-mediated genome editing system in memory Th2 cells (Supplementary Fig. 3a). As shown in Fig. 3c, d, we checked the genome sequence of *Dusp10* or protein levels of DUSP10 in edited Tpath2 cells and confirmed that the usage of single-guide RNA for *Dusp10* effectively resulted in the genome editing of *Dusp10*. We found that CRISPR/Cas9-mediated genetic deletion of *Dusp10* enhanced the production of IL-5 and IL-13 by Tpath2 cells in response to IL-7 plus IL-33 (Fig. 3e, $n = 3$ for each group, *$p < 0.05$). We confirmed the upregulation of *Il5* and *Il13* mRNA

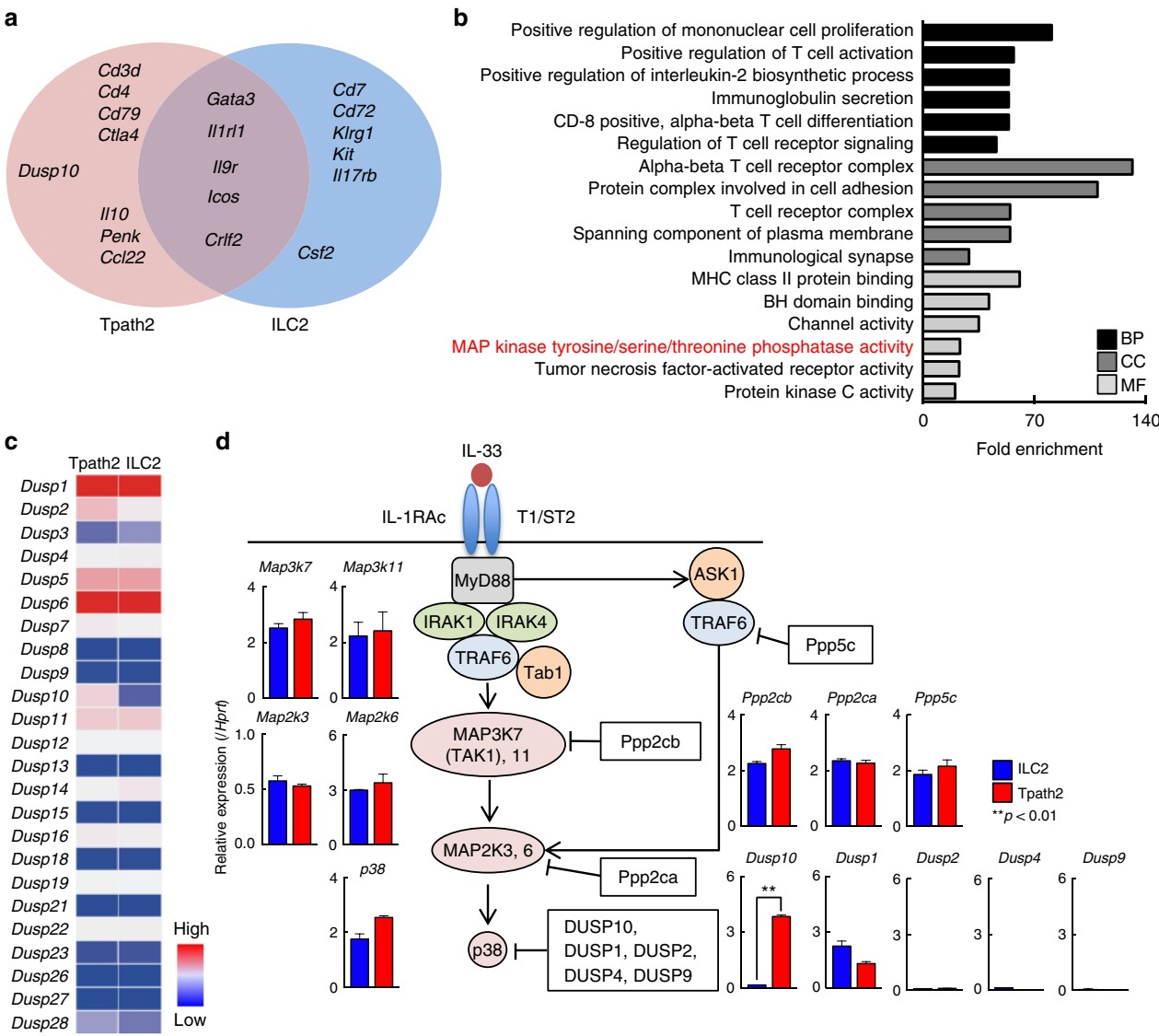

**Fig. 2** Preferential expression of *Dusp10* in Tpath2 cells. **a–d** Resting Tpath2 cells and ILC2s were used in RNA-sequencing. Cells isolated from spleens of memory Th2 mice (as shown in Supplementary Fig. 1a). After cultivation with IL-7, IL-25 plus IL-33 for 5 days, cells were rested with IL-7 alone for 24 h and analyzed by RNA-sequencing. **a** Venn diagram shows the genes preferentially expressed in Tpath2 cells and ILC2s. Immune response-related genes (GO0006955) preferentially expressed by unstimulated Tpath2 cells (Red) or unstimulated ILC2s (Blue) (>5-fold difference), or genes shared by the two (Purple) (<1.2-fold difference) are shown. Transcripts expressed above 10 fragments per kilobase of exon per million reads mapped (FPKM) are shown. **b** Gene Ontology analysis of genes that are expressed more highly in Tpath2 cells as compared to ILC2s. Graph shows folds change of indicated GO terms based on sub-categories including biological process (BP), cellular component (CC), and molecular function (MF) in Tpath2 cells to ILC2s (False discovery rate < 0.05). *X*-axis shows the number of fold-change ($p < 0.02$, fold change of at least 10, BP biological process, CC cellular component, MF molecular function). Transcripts expressed above 10 FPKM in Tpath2 cells were used. **c** A heatmap of genes distinctly expressed between Tpath2 cells and ILC2s in the category of MAP kinase phosphatase activity as in **b**. **d** Quantitative RT-PCR analysis of the kinases and phosphatases related to IL-33–ST2 signaling pathway in Tpath2 cells and ILC2s. Two independent experiments were performed and showed similar results (**a–c**). More than three independent experiments were performed and showed similar results (**d**) (\*\*$p < 0.01$). Three technical replicates were performed with quantitative RT-PCR (**d**)

expression in *Dusp10*-deficient Tpath2 cells in response to IL-7 plus IL-33 (Fig. 3f). The protein levels of IL-5 and IL-13 were augmented in *Dusp10*-deficient Tpath2 cells in response to IL-7 plus IL-33 (Fig. 3g). We also addressed the role of DUSP10 in IL-7/33-induced p38 phosphorylation in Tpath2 cells, and enhanced levels of phospho-p38 was detected following deletion of *Dusp10* (Fig. 3h, $n = 4$ for each group, \*$p < 0.05$). Furthermore, we induced over-expression of *Dusp10* in Tpath2 cells using a retrovirus gene introduction system, and as expected, this reduced the number of IL-7/33-induced IL-5$^+$ memory Th2 cells (Supplementary Fig. 3b and c). Thus, these results indicate that DUSP10 negatively regulates the phosphorylation of p38 MAPK

leading to the inhibition of IL-33-induced IL-5 and IL-13 production in Tpath2 cells.

**Induction of *Dusp10* suppresses IL-5 production in ILC2.** Next, we tested whether expression of *Dusp10* affects IL-33-induced Th2 cytokine production in ILC2s. ILC2s were infected with a retrovirus containing the human *NGFR* and *Dusp10*, and IL-7/33-induced cytokine production was assessed (Fig. 4a). The expression of human *NGFR* reflects the efficiency of gene transduction (Fig. 4b and Supplementary Fig. 4a). It has been reported that a single amino acid change (Cysteine to Serine) in the phosphatase

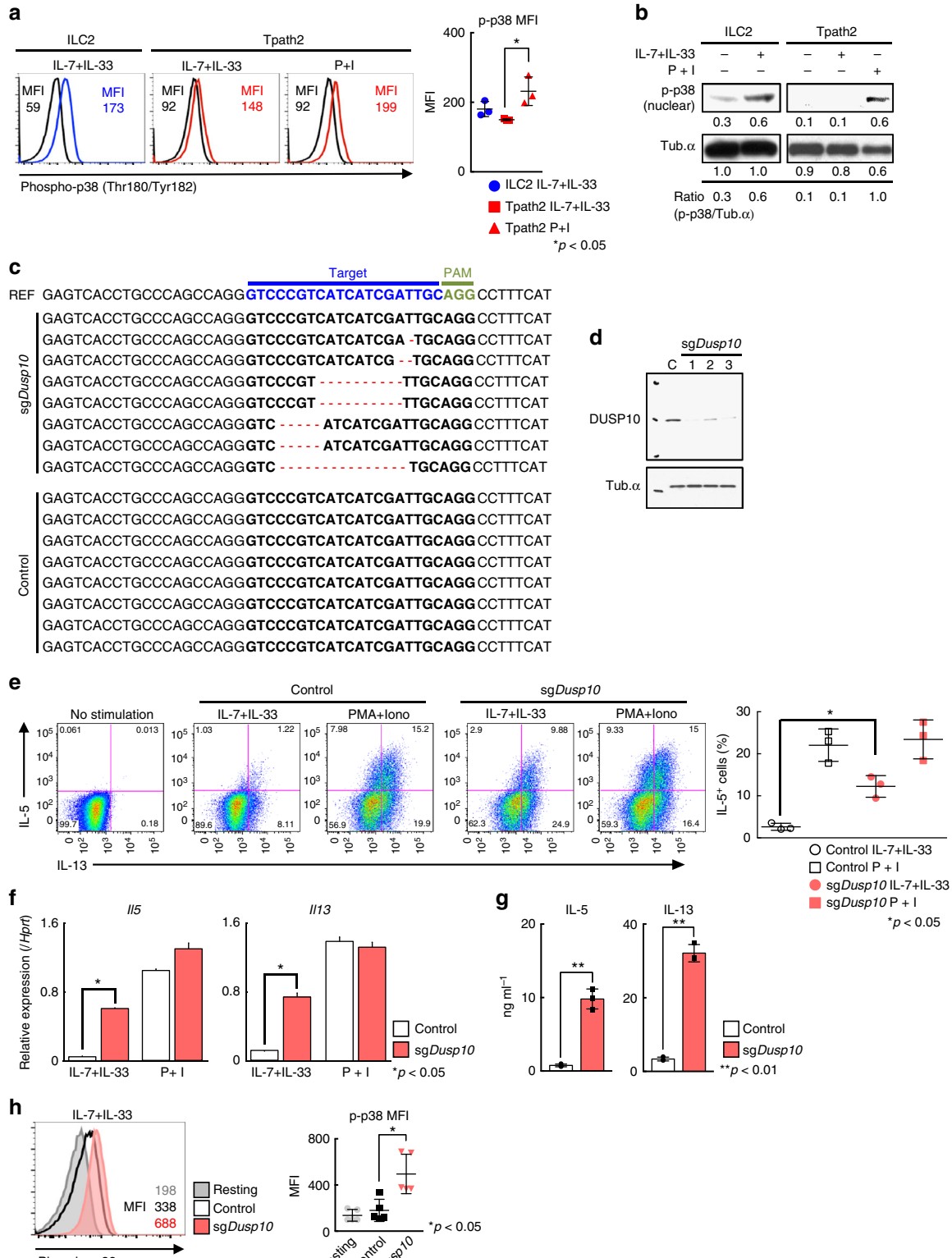

domain of DUSP10 abrogated its phosphatase activity, and therefore we also included this mutant (C409S) form of DUSP10 in the assays[38] (Fig. 4a–c). We confirmed that the levels of cell surface and mRNA expression of *Il1rl1* in ILC2s were not dramatically changed by enforced expression of *Dusp10* (Supplementary Fig. 4b and c). Enforced expression of *Dusp10* in ILC2s suppressed the production of IL-5 and IL-13 to similar levels as that observed in ILC2s treated with a p38 inhibitor, SB203580

(Fig. 4c, n = 3 for each group, *p < 0.05). Interestingly, the *Dusp10* C409S mutant did not affect cytokine production from ILC2s (Fig. 4c, far right). Similarly, mRNA expression of *Il5* and *Il13* in *Dusp10*-overexpressing ILC2s after IL-33 stimulation were significantly lower than the levels detected in control and *Dusp10* C409S mutant groups (Fig. 4d). Protein expression levels of IL-5 and IL-13 were also assessed by ELISA, and similar results were obtained (Fig. 4e). We further demonstrated the effect of *Dusp10*

**Fig. 3** Deletion of *Dusp10* increases IL-5 production in Tpath2 cells. **a** Intracellular staining profiles of phospho-p38 in Tpath2 cells and ILC2s purified from memory Th2 mice (as shown in Supplementary Fig. 1a) after stimulation with PMA plus Ionomycin (P + I) or IL-7 plus IL-33 (IL-7 + IL-33) for 30 min. The number in the histogram represents MFI. Black line shows non-stimulated cells. Graph shows mean MFI ± SD values of phospho-p38 in stimulated cells ($n$ = 3, separately). **b** Phosphorylated p38 (p-p38) and Tub.α in Tpath2 cells and ILC2s after stimulation with P + I, or IL-7 + IL-33 for 30 min. Band intensities were measured with a densitometer. The arbitrary ratio of the intensity of p-p38 to that of tubulin is shown. **c** Deletion patterns of the *Dusp10* locus in control Tpath2 cells and sg*Dusp10*-treated Tpath2 cells. Reference (REF) sequence is shown on top of clonal sequences from each population with sgRNA target (blue) and PAM (green) sequences indicated. Red dashes denote deleted bases. **d** DUSP10 in sg*Dusp10*-treated Tpath2 cells. C refers to Cas9 Ctrl Tpath2 cells. An antibody to tubulin (Tub) was used as a loading control. **e**) Intracellular staining profiles of IL-5 and IL-13 in sg*Dusp10*-treated Tpath2 cells (sg*Dusp10*) and non-targeted Tpath2 cells (Control) are shown. The cells were stimulated with indicated stimuli for 6 h. The percentages of IL-5 positive Tpath2 cells are shown (mean ± SD; $n$ = 3 for each group). **f** Quantitative RT-PCR analysis of *Il5* and *Il13* in edited cells and control cells after indicated stimulation for 4 h. Relative expression (normalized to *Hprt*) is shown with standard deviation. **g** ELISA analysis of the indicated cytokines from Tpath2 cells. $1 \times 10^4$ cells were stimulated with indicated stimuli for 48 h. **h** Intracellular staining of phospho-p38 in *Dusp10* edited cells and control after stimulation with IL-7 plus IL-33 for 30 min. Gray filled histogram shows isotype control staining. Graph shows mean MFI ± SD values of phospho-p38 in each edited cells ($n$ = 5, separately). Three independent experiments were performed and showed similar results (**a–h**) (**$p < 0.01$, *$p < 0.05$). Three technical replicates were included in **f** and **g**

on the phosphorylation status of p38 in ILC2s. Phosphorylation of p38 was induced by IL-33 plus IL-7 stimulation in ILC2s (Fig. 4f, upper panel). The levels of phospho-p38 were lower in *Dusp10*-overexpressing ILC2s as compared to Mock (empty vector)-infected ILC2s with or without IL-7 plus IL-33 stimulation (Fig. 4f, g). Thus, these results indicate that DUSP10 negatively regulates the phosphorylation of p38 MAPK leading to the inhibition of IL-33-induced IL-5 and IL-13 production in ILC2s.

**DUSP10 suppresses the transcriptional activity of GATA3.** GATA3 is known to bind to the *Il5* promoter and control its activity in memory-type pathogenic Th2 cells[22]. In addition, phosphorylation of GATA3 by p38 is critical for Th2 cytokine production by Th2 cells and ILC2s[28,39]. To identify the molecular mechanism by which DUSP10 negatively controls IL-33-induced IL-5 and IL-13 production via the inhibition of p38 phosphorylation, we sought to demonstrate the activation of GATA3 and possible physical association of GATA3 with p38 in the presence or absence of DUSP10. Physical association of GATA3 with p38 was easily detected in the precipitates from 293T cells transfected with GATA3 (Fig. 5a, lane 2), and the DUSP10 molecule was also associated with p38 (Fig. 5a, lanes 3 and 4). However, the association between GATA3 and p38 was abrogated in the presence of DUSP10 (Fig. 5a, lane 4). The association among DUSP10, GATA3, and p38 appeared to not be dependent on IL-33, since IL-33 was not added to the cells. Having shown that DUSP10 and p38 can interact, we next assessed the effect of DUSP10-overexpression on the levels of phospho-GATA3 and phospho-p38. While phospho-GATA3 was easily detected together with phosphorylation of p38 in the absence of DUSP10 (Fig. 5b, lane 2), the levels of phospho-GATA3 and phoshpo-p38 were dramatically reduced in the presence of DUSP10 (Fig. 5b, lanes 3 and 4). Total amounts of GATA3 and p38 were not changed in the presence or absence of DUSP10 (Fig. 5b). These results indicate that DUSP10 suppresses phosphorylation of GATA3 through the inhibition of p38 phosphorylation. We also examined the phosphorylation of GATA3 in Tpath2 cells and ILC2s. We detected strong induction of phospho-GATA3 in the nuclear fraction of Tpath2 cells following stimulation with PMA and Ionomycin (Fig. 5c, right), while the levels were moderate after stimulation with IL-7 plus IL-33 (Fig. 5c, middle). Phospho-GATA3 was easily detected in ILC2s stimulated with IL-7 plus IL-33 (Fig. 5d, left), while the phopho-GATA3 was decreased by the transduction of *Dusp10* under IL-7/33 stimulated conditions (Fig. 5d, right). These results indicate that DUSP10 plays a critical role in the inhibition of IL-33-induced phosphorylation of GATA3 in primary cells.

To assess the effect of DUSP10 on the DNA-binding activity of GATA3, a pull-down assay with the *Il5*p sequence was performed. In the presence of DUSP10, the binding of GATA3 to the *Il5*p was substantially decreased in a dose-dependent manner (Fig. 5e). We also performed the DNA pull-down assay by adding a mutant oligonucleotide (a mutant *Il5*p oligonucleotide that GATA3 cannot bind) as a negative control (Supplementary Fig. 5). Furthermore, the suppressive effect of DUSP10 on *Il5*p activity was assessed. As expected, *Dusp10* WT efficiently suppressed GATA3-induced *Il5*p activity, however, the *Dusp10* C409S mutant did not (Fig. 5f). To test the effect of DUSP10 on GATA3 binding to the Th2-cytokine gene loci in ILC2s, we next performed a ChIP assay with *Dusp10*-overexpressing ILC2s (Fig. 5g). At the *Il5* locus, the binding of GATA3 to the *Il5*p and *Il5* U1 region was reduced by the enforced expression of *Dusp10* in ILC2s (Fig. 5g, top panel, $n$ = 4, *$p < 0.05$, **$p < 0.01$). In addition, GATA3 binding to the *CGRE* and *Il13*p was reduced by the induction of *Dusp10* in ILC2s (Fig. 5g, bottom panel, $n$ = 4, *$p < 0.05$, **$p < 0.01$). These results indicate that DUSP10 inhibited the binding of GATA3 to the *Il5*p and *Il13*p and reduced the ability of GATA3 to transcriptionally activate the *Il5*p.

**DUSP10 controls IL-33-induced airway inflammation in vivo.** In order to investigate the physiological consequences of DUSP10-mediated regulation of IL-33-induced IL-5 production, we used an ILC2-dependent papain-induced airway inflammation model, as illustrated in Supplementary Fig. 6a–c. ILC2s were purified from Thy1.1$^+$ BLAB/c mice after IL-33 administration for consecutive 5 days, and Mock- or *Dusp10*-containing retrovirus were transduced. Retrovirally-transduced *Dusp10*-expressing ILC2s were then purified by cell sorting and were transferred into Thy1.2$^+$ BALB/c *nu/nu* mice. In order to deplete intrinsic ILC2s in BALB/c *nu/nu* recipient mice, we administrated Thy1.2 antibody on days 0, 1, and 3. We examined the effect of the depletion of ILC2s on papain-administered mice, in which Thy1.2$^+$ILC2s were depleted by the administration of Thy1.2 antibody (62.4 to 2.3% in Lin$^-$CD127$^+$ population in the lung) (Supplementary Fig. 6b). We also assessed bronchoalveolar lavage (BAL) fluid 24 h after the last papain administration, and the numbers of infiltrated eosinophils in the BAL fluid were found to be almost completely abrogated by the depletion of ILC2s (Supplementary Fig. 6c). Thus, in this system, the contribution of host Thy1.2$^+$ILC2s to the eosinophilic inflammation was very low. Using this system, we found that the group that received *Dusp10*-overexpressing ILC2 (*Dusp10*-ILC2s) cells showed a significant decrease in the number of inflammatory eosinophils in the BAL

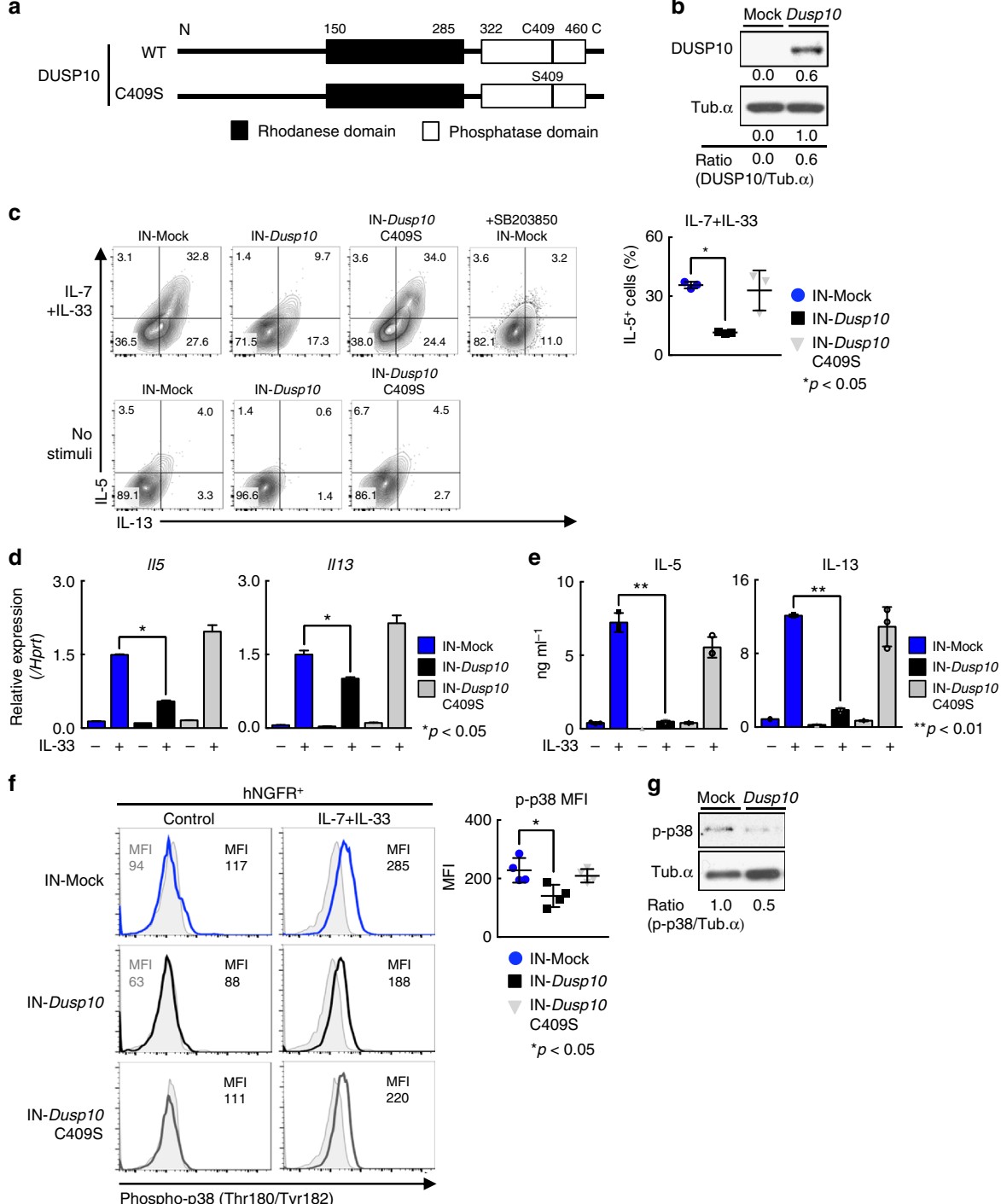

**Fig. 4** Induction of *Dusp10* suppresses IL-5 production in ILC2. **a** Schematic representation of wild-type *Dusp10* (WT) and C409S mutant. **b–g** Freshly prepared ILC2s purified form the spleen of IL-33-injected BALB/c mice were infected with a Mock (empty vector), *Dusp10*, or C409S mutant-IRES-hNGFR-containing retrovirus and then cultured with IL-7, IL-25 and IL-33. 5 days after infection, hNGFR-positive ILC2 cells were enriched by cell sorting. **b** Western blot analysis of DUSP10 in freshly prepared ILC2s and *Dusp10*-overexpressing ILC2s. The arbitrary ratio of the intensity of DUSP10 to that of tubulin is shown. **c** Freshly prepared ILC2s were infected with a *Dusp10*, or C409S mutant-IRES-hNGFR-containing retrovirus and then cultured with IL-7, IL-25, and IL-33. 5 days after infection, intracellular-staining profiles of IL-5 and IL-13 in hNGFR+ ILC2s are shown. Cells were purified and then stimulated with IL-7 plus IL-33 for 6 h with or without 30 min treatment of SB203850 before stimulation. The percentages of IL-5 positive cells in hNGFR+ ILC2s were shown (mean ± SD; *n* = 3 for each groups). **d** Quantitative RT-PCR analysis of *Il5* and *Il13* in hNGFR+ ILC2s. Cells were stimulated with IL-7 plus IL-33 for 4 h. **e** ELISA analysis of the culture supernatant from $3 \times 10^4$ cells of hNGFR+ ILC2s. Those cells were sorted as in **d**, and then stimulated with IL-7/33 for 12 h. **f** Intracellular-staining of phospho-p38 of hNGFR+ ILC2s after stimulation with IL-7 + IL-33 for 30 min. Mock represents empty vector introduction. Graph shows mean MFI ± SD values of phospho-p38 in hNGFR+ ILC2s (*n* = 4), separately. **g** Phosphorylated p38 (p-p38) in freshly prepared ILC2s with *Dusp10*-overexpression after stimulation with IL-7 + IL-33 for 30 min. The arbitrary ratio of the intensity of p-p38 to that of tubulin is shown. More than four independent experiments were performed and showed similar results (**b–g**) (**p < 0.01; *p < 0.05; Mann–Whitney *U* test; **b–g**). **d**, **f**, **g** Three technical replicates were performed with quantitative RT-PCR and ELISA analysis

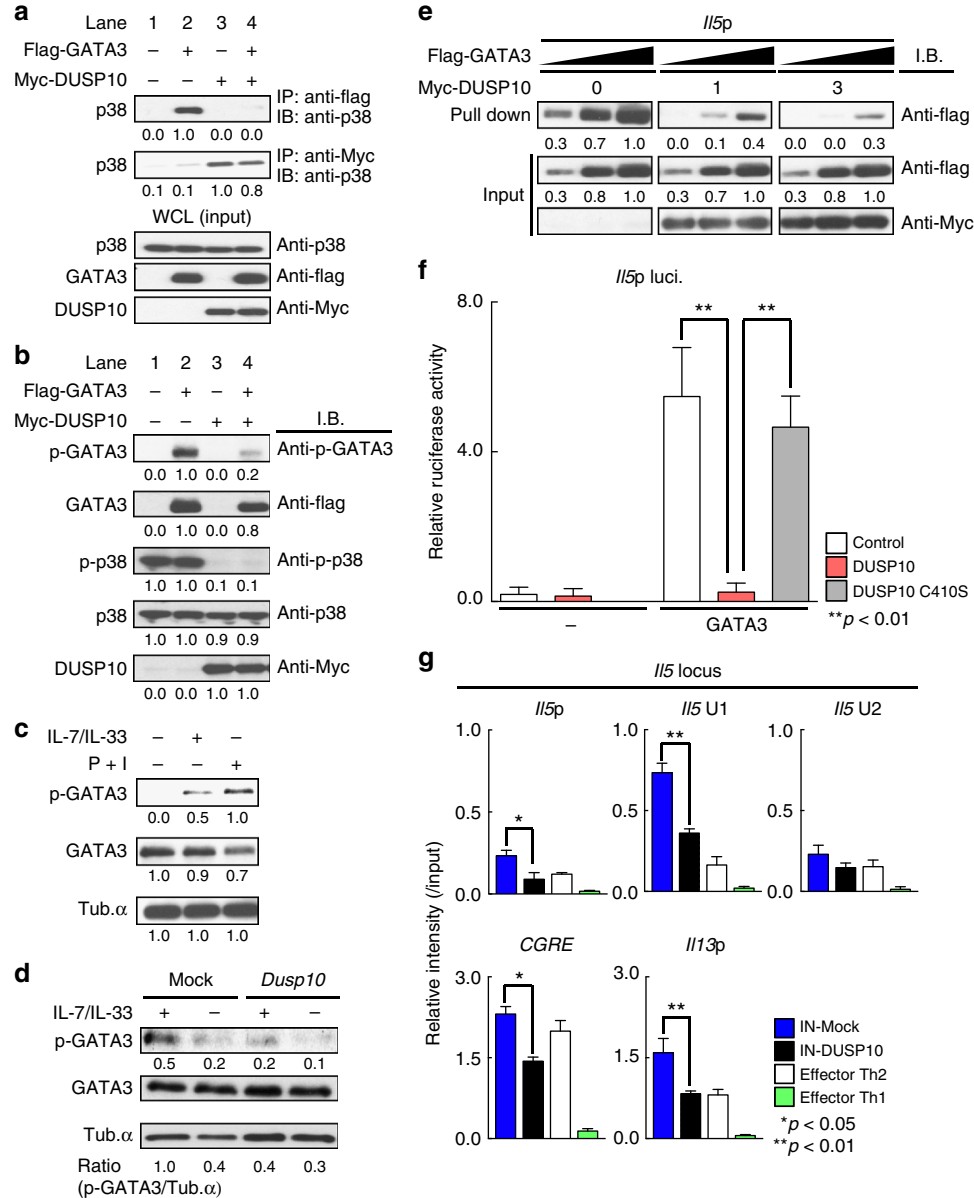

**Fig. 5** DUSP10 suppresses the transcriptional activity of GATA3. **a, b** 293T cells were transfected with Myc-tagged DUSP10 or Flag-tagged GATA3. **a** Immunoprecipitation assay was performed with anti-Myc or anti-Flag. Immunoblotting of whole-cell lysates is also shown as a control (Input). **b** The phosphorylated GATA3 (p-GATA3), total GATA3 (GATA3), phosphorylated p38 (p-p38), total p38 (p38) and total DUSP10 (DUSP10) in whole fraction. **c** The phosphorylated GATA3 (p-GATA3) and total GATA3 (GATA3) in the nuclear fraction from Tpath2 cells. Tpath2 cells were stimulated with P + I, or IL-7 + IL-33 for 30 min. **d** The phosphorylated GATA3 (p-GATA3) and total GATA3 (GATA3) in the nuclear fraction from *Dusp10*-overexpressing ILC2s shown in Fig. 4b. ILC2s were stimulated with IL-7 + IL-33 for 30 min. The arbitrary ratio of the intensity of p-GATA3 to that of tubulin was shown. **e** 293 T cells were transfected with Myc-tagged DUSP10 or Flag-tagged GATA3, and total extracts were subjected to a pull-down assay using *Il5* promoter oligonucleotide as described in the Supplementary information. Immunoblotting of total cell lysates is also shown (Input). The *x*-axis of GATA3 refers to the amounts of cell lysates (3-fold doses; DNA-precipitants and input samples) that were blotted with anti-Flag Ab. Band intensities were measured with a densitometer and arbitrary densitometric units are shown (**a–e**). **f** Reporter assays with the *Il5* promoter were performed using the *Dusp10*-transfected D10. G4.1 Th2 cell line. The mean values with standard deviations of relative luciferase activity of three different experiments are shown. **g** A ChIP assay was performed with anti-GATA3 at the *Il5* gene locus, *CGRE* and *Il13p* in Control- or *Dusp10*-overexpressing ILC2s as shown in Fig. 4b, and effector Th1 cells and effector Th2 cells. **a–g** Three independent experiments were performed with similar results (**p < 0.01; *p < 0.05; Mann–Whitney U test; **f, g**). Three technical replicates were performed for reporter (**f**) and ChIP (**g**) assays

fluid compared to the group that received non-transduced ILC2 (Mock-ILC2s) cells (Fig. 6a, n = 5 for each group, **p < 0.01). Histological analysis revealed a similar reduction in mononuclear cell infiltration into the peribronchiolar regions of the lungs (Fig. 6b). Periodic acid-Schiff (PAS) staining also showed decreased production of mucus in the *Dusp10*-ILC2s group compared to the Mock-ILC2s group (Fig. 6c). Hence, *Dusp10*-

overexpressing ILC2s were rendered unable to induce eosino-philic airway inflammation induced by papain administration.

Papain is known to induce IL-33 release and also promote production of other tissue checkpoint cytokines including TSLP[18]. To more directly evaluate the role of DUSP10 in IL-33-induced airway inflammation, we next administrated IL-33 directly to the mice. Intranasal administration of IL-33 induces

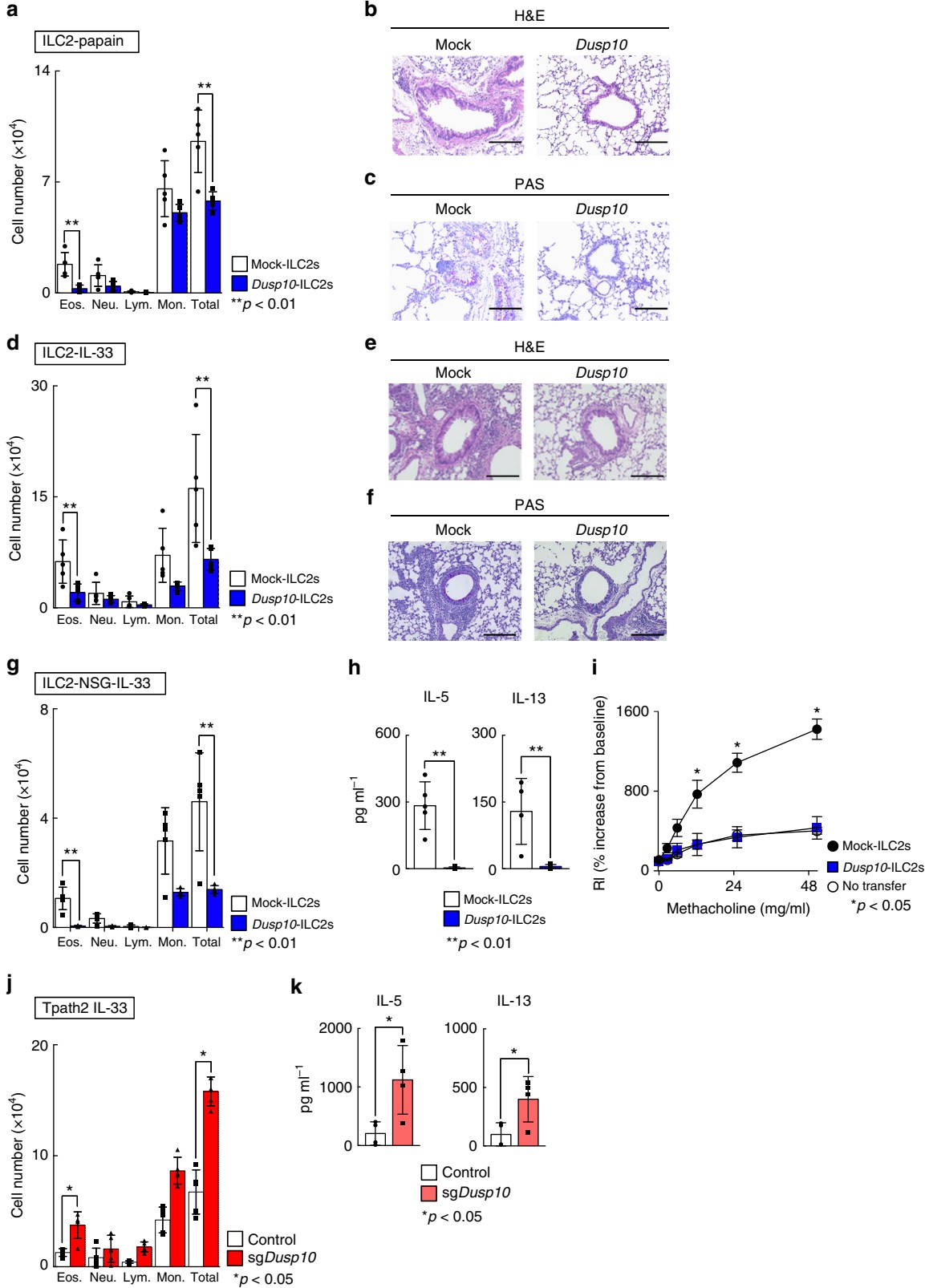

acute airway eosinophilic inflammation independent of T and B cells that is largely dependent on innate cells including ILC2s[40]. To assess whether inflammation induced by IL-33 administration could be reduced by the transduction of *Dusp10* into ILC2s, we next transferred *Dusp10*-ILC2s into BALB/c *nu/nu* mice, similar to Supplementary Fig. 6a, followed by administration of IL-33 to

the mice intranasally for 3 consecutive days (Supplementary Fig. 6d). As expected, the *Dusp10*-ILC2 group showed a significant decrease in the number of inflammatory eosinophils in the BAL fluid as compared to the Mock-ILC2s group (Fig. 6d, n = 5 for each group, **p < 0.01). Similarly, histological analysis of lung tissue showed a considerable reduction in mononuclear

**Fig. 6** DUSP10 controls IL-33-induced airway inflammation in vivo. **a** The absolute cell numbers of eosinophils (Eos), neutrophils (Neu), lymphocytes (Lym), and monocytes (Mono) in the BAL fluid collected from mice treated with papain as described in Supplementary Fig. 6d are shown. Mean values (five mice per group) are shown with standard deviations. **b**, **c** Lung tissue sections were fixed and stained with H&E (**b**) or PAS (**c**). A representative staining pattern is shown. Scale bars represent 100 μm. **d** The absolute numbers of leukocytes in the BAL fluid collected from mice treated with rmIL-33 as described in Supplementary Fig. 6d are shown. Samples were collected 24 h after the last intranasal IL-33 administration. Mean values (five mice per group) are shown with standard deviations. **e**, **f** Lung tissue sections were fixed and stained with H&E or PAS. A representative staining pattern is shown. Scale bars represent 100 μm. **g** The absolute numbers of leukocytes in the BAL fluid collected from NSG mice treated with rmIL-33 as described in Supplementary Fig. 6e are shown. **h** ELISA of the indicated cytokines in the BAL fluid from each experimental group shown in **g**. **i** Lung resistance (RL) was assessed in response to increasing doses of methacholine. The mean values (four mice in Mock-ILC2s and No transfer group, five mice in *Dusp10*-ILC2s transfer group, respectively) are shown with SDs. **j** The absolute numbers of leukocytes in the BAL fluid collected from mice transferred with *Dusp10*-deficient Tpath2 cells and then treated with rmIL-33 as described in Supplementary Fig. 6g are shown. **k** ELISA of the indicated cytokines in the BAL fluid from each experimental group shown in **j**. Two independent experiments were performed and showed similar results (**$p < 0.01$; *$p < 0.05$; Mann–Whitney U test). Three technical replicates were included in ELISA on the BAL fluid samples

cell infiltration into the peribronchiolar regions, as was observed during papain-induced inflammation (Fig. 6e). PAS staining also showed decreased mucus production in the *Dusp10*-ILC2 group (Fig. 6f). We further transferred *Dusp10*-ILC2s into NOD.Cg-*Prkdc*[scid]*Il2rg*[tm1Wjl]/SzJ (NSG) mice, which lack NK cells, T cells, B cells and ILCs (Supplementary Fig. 6e). Significantly decreased infiltration of eosinophils into the BAL fluid (Fig. 6g, $n = 5$ for each group, **$p < 0.01$), and decreased IL-5 and IL-13 production in the BAL fluid (Fig. 6h, $n = 5$ for each group, **$p < 0.01$) were observed in the *Dusp10*-ILC2 group compared Mock-ILC2s group (Fig. 6g, h). We also analyzed the degree of AHR in the allergy-induced mice, which received Mock- or *Dusp10*-ILC2s, by measuring methacholine-induced airflow obstruction with a mechanical ventilator. The degree of AHR in mice that received *Dusp10*-ILC2s was lower than that of mice that received Mock-ILC2s (Fig. 6i, $n = 4$ or 5 for each group, *$p < 0.05$). Consistent with these observations, *Muc5ac* mRNA expression in the lung tissue was also decreased in mice that received *Dusp10*-ILC2s without affection the expression levels of *Gata3* and *Il1rl1* (Supplementary Fig. 6f). Thus, IL-33-induced ILC2-dependent eosinophilic inflammation is ameliorated by the transduction of *Dusp10*.

Finally, we further examined the effect of CRISPR/Cas9-mediated genome editing of *Dusp10* on Tpath2 cell function and the ability of Tpath2 cells to induce IL-33-induced eosinophilic airway inflammation in vivo. We generated *Dusp10*-deficient Tpath2 cells using the CRISPR/Cas9 system as illustrated in Supplementary Fig. 6g. *Dusp10*-deficient Tpath2 cells were transferred intravenously into NSG mice. We then administered IL-33 to these mice intranasally and monitored development of airway inflammation after transfer of *Dusp10*-sufficient and *Dusp10*-deficient Tpath2 cells. Cas9-mediated knockout of *Dusp10* in Tpath2 cells resulted in a significant increase in the number of inflammatory eosinophils in the BAL fluid as compared to the control group (Fig. 6j, k, $n = 5$ for each group, *$p < 0.05$). Taken together, these in vivo results demonstrate that DUSP10 is responsible for the differential ability of ILC2 and Tpath2 cells to directly respond to IL-33. Moreover, DUSP10 constrains IL-33-induced eosinophilic airway inflammation by modulating cytokine production by Tpath2 cells and ILC2s.

## Discussion

Here, we have identified a mechanism that regulates a fundamental difference between the innate and adaptive immune system. Acquired, or adaptive immunity depends on T cells that require recognition of cognate antigen via their TCR to induce cytokine production and effector function. In contrast, innate lymphoid cells can be activated and can produce effector cytokines in response to direct exposure to signals such as IL-33. We show that Dual-specificity phosphatase (DUSP) 10, a MAP kinase

phosphatase, is a central regulatory molecule that is responsible for the inability of adaptive T cells to produce cytokines in an innate-like fashion. *Dusp10* is highly expressed in Tpath2 cells but expression is very low in ILC2s. DUSP10 inhibits IL-33-induced p38 MAPK phosphorylation resulting in impaired GATA3 phosphorylation and GATA3-mediated transcriptional upregulation of Th2 cytokines. These findings may explain why Tpath2 cells do not produce IL-5 and IL-13 in response to IL-33 even though they express high levels of IL-33 receptor ST2. Genome editing of *Dusp10* resulted in increased production of IL-5 and IL-13 in Tpath2-like cells after stimulation with IL-33. In contrast, enforced expression of *Dusp10* in ILC2s inhibited IL-33-induced IL-5 and IL-13 production. Furthermore, deletion of *Dusp10* in Tpath2 cells or forced expression of *Dusp10* in ILC2s resulted in exacerbated or attenuated allergic responses in the airway, respectively. Thus, DUSP10 constrains IL-33-induced Th2 cytokine production by Tpath2 cells and ILC2s via the inhibition of the p38-GATA3 activation pathway, and regulating DUSP10 activity could represent a new therapeutic target for the treatment of allergic inflammation induced by Tpath2 cells and ILC2s.

In the current study, the expression of several DUSPs was detected both in Tpath2 cells and ILC2s, which indicated that these DUSPs may participate in the function of these cells. Among them, only *Dusp10* and *Dusp2* were differentially expressed between these two cell types. Given the large number of DUSPs that are often co-expressed in one cell type, redundancy and functional compensation could be expected when just a single DUSP gene is inactivated[37]. In fact, genetic deletion of *Dusp10* in Tpath2 cells did not completely rescue the induction of IL-33-induced IL-5 production as to the levels induced by PMA plus Ionomycin. The substrate specificity of DUSPs should be considered when assessing possible redundant and compensatory functions for DUSP family members[33,41]. JNK and p38 MAPK appear to be the preferred substrates of DUSP10 in vitro[37]. Similarly, both DUSP1 and DUSP2 dephosphorylate p38 MAPK in vitro. In the current study, however substantial rescue effects observed by the deletion of *Dusp10* indicate that DUSP10 is the major regulator of IL-33-mediated p38 MAPK phosphorylation, leading to IL-5 and IL-13 expression in Tpath2 cells and ILC2s.

Recent studies highlight a unique role of phosphatases including DUSPs in the function of various immune cells[42–44]. DUSP2 dephosphorylated the transcriptional activator STAT3 during Th17 cell differentiation and limited the development of Th17 cells, suggesting that DUSP2 modulates susceptibility to experimental colitis[44]. However, pro-inflammatory roles for DUSP2 were also reported in macrophages[44,45]. Zhang and colleagues have also reported that *Dusp10*-deficient macrophage exhibit enhanced inflammatory cytokine production upon TLR stimulation[46].

GATA3 is essential for Th2-cytokine production and controls the expression of various genes in Th2 cells and ILC2s[19,34]. The p38 MAPK phosphorylates GATA3 to induce translocation to the nuclear compartment in human Th2 cells[39]. IL-33-mediated production of IL-5 and IL-13 in ILC2s and IL-33-mediated exacerbation of pathogenicity of Tpath2 cells depend on the phosphorylation of p38 and GATA3[23,28]. In the current study, we found that DUSP10 physically interacts with and dephosphorylates p38 MAPK, via the residues Thr180 and Tyr 182, resulting in the inhibition of GATA3 phosphorylation and its binding to the *Il5* and *Il13* genes. Interestingly, although *Dusp10* is substantially expressed in Tpath2 cells, the IL-33-p38 signaling pathway could induce chromatin remodeling at the *Il5* gene locus in Tpath2 cells. In fact, permissive chromatin conformation at the *Il5* gene locus was detected in Tpath2 cells at equivalent levels to ILC2s. Thus, different molecular machinery may organize the IL-33-mediated GATA3-dependent chromatin remodeling and GATA3-induced transactivation of Th2 cytokines. Similar to these observations, we have reported that methylation of GATA3 at Arg261 has selective defect in the transactivation of *Il5* without affecting chromatin status[47]. In another case, Akt-1-mediated GATA3 phosphorylation repressed IFNγ production by memory Th2 cells[48]. Thus, the functional diversity of GATA3 is precisely controlled by posttranslational mechanisms[19]. In the current study, we demonstrate that DUSP10 constrains IL-33-induced Th2 cytokine production by Tpath2 cells via the inhibition of p38-mediated posttranslational modification of GATA3.

The results from our mouse airway inflammation models indicate that DUSP10 functions as a negative regulator of IL-33-mediated allergic airway inflammation by limiting Th2 cytokine production in Tpath2 cells in response to IL-33. Recently, global transcriptional profiling to determine pathogenic features associated with asthma and rhinitis has been reported by analyzing Th2 cells from subjects with allergic asthma and rhinitis[49]. They found significantly lower levels of *DUSP10* in Th2 cells from asthmatic subjects. Thus, it is conceivable that the decreased expression of *DUSP10* in Th2 cells causes allergic diseases and/or is associated with the pathogenesis of chronic allergic disorders.

Interestingly, both ILC1s and ILC3s express high levels of *Dusp10* as compared to ILC2s. However, ILC1s and ILC3s do not express the IL-33 receptor ST2[24]. Reactivity to IL-33 in ILCs is therefore controlled by at least two distinct mechanisms; ST2 expression and *Dusp10* expression. Conventional effector Th2 cells express low levels of *Dusp10*, and limited amounts of ST2[23], while Tpath2 cells express high levels of *Dusp10* and ST2. Thus, Tpath2 cells may acquire increased expression of ST2 and also high expression of *Dusp10* during their differentiation from effector Th2 cells. In any event, reactivity to IL-33 in Tpath2 cells appears to be strictly regulated, particularly in the IL-33-abundant environment of the inflamed airway[10]. Indeed, it is known that mucosal barriers such as the airway are continuously exposed to a variety of microorganisms and allergens, and undergo a continuous process of damage and repair. The release of epithelial cell-derived cytokines, such as IL-33 therefore likely occurs daily. Strict negative regulation of reactivity to IL-33 in antigen-specific helper T cells is therefore beneficial.

We identified IL-5-producing ST2[hi] pathogenic Th2 cells (Tpath2 cells) that likely induce eosinophilic inflammation in nasal polyps of Eosinophilic Chronic Rhinosinusitis (ECRS) patients[19,23]. The antigen-specificity of these cells remains unknown. Similar pathogenic Th2 cell populations with a hPGDS[hi]CRTH2[hi] phenotype was identified in the peripheral blood of patients with eosinophilic gastrointestinal diseases and atopic dermatitis[50]. More recently, Wambre et al. identified allergen-specific human pathogenic Th2 cells with a CRTH2[hi]CD161[hi] phenotype that play a role in the pathogenesis

of various allergen-specific atopic diseases[51]. Based on their observations, antigen-induced TCR-mediated activation of the pathogenic Th2 cells appeared to be critical for the pathogenesis of eosinophilic inflammation in these allergic patients. Although further investigation is necessary, DUSP10 may have an important function as a negative regulator of Th2 cytokine production in pathogenic Tpath2 cells in allergic patients.

As discussed above, *Dusp10* expression in Th2 cells has been associated with the pathogenesis of allergic disorders including asthma and rhinitis[49]. Therefore, small molecules that upregulate DUSP10 or mimic DUSP10 function could be used as new therapeutic agents for the treatment of allergic diseases. Vitamin D receptor (Vdr), a nuclear receptor transcription factor, was shown to control *Dusp10* expression in human adipocyte and prostate cells[52,53]. Moreover, 1,25 $(OH)_2D_3$, a hormonally active metabolite of Vitamin D, inhibits in vitro polarization of primary murine $CD4^+$ T cells towards Th2 cell differentiation[54]. Furthermore, a meta-analysis revealed that the exacerbation of asthma appeared to be attenuated by the internal use of 1,25 $(OH)_2D_3$[55]. Thus, although further studies focus on the role of Vdr in the expression of DUSP10 in Tpath2 cells and ILC2s are required, 1,25 $(OH)_2D_3$ could be involved in the control of *Dusp10* expression, which may lead to the amelioration of the pathology of asthma.

In summary, we demonstrate a mechanism that differentiates the major innate characteristic of ILC2s, cytokine-induced cytokine expression, from antigen-dependent adaptive Th2 cell function. We show that DUSP10, a MAP kinase phosphatase plays a critical role suppressing the innate function of T cells. DUSP10 is highly expressed in ST2[+] Tpath2 cells and negatively controls the ability of these cells to directly respond to IL-33 and produce Th2 cytokines. This is accomplished by inhibition of the p38 MAPK-GATA3 signal axis. Thus, *Dusp10* expression, DUSP10-mediated regulation of p38 MAPK activation and resulting phosphorylation of GATA3 in these cells could be potential therapeutic targets for the treatment of intractable allergic disorders.

## Methods

**Mice**. OVA-specific TCR-αβ (DO11.10) transgenic (Tg) mice were provided by Dr. D. Loh (Washington University School of Medicine, St. Louis), and backcrossed to BALB/c mice ten times[56]. BALB/c, BALB/c *nu/nu* and C57BL/6 mice were purchased from CLEA Japan. Thy1.1 mice with a BALB/c background were purchased from Jackson Laboratories. NSG mice were purchased from Charles river laboratories. All mice were used at 6–8 weeks old and were maintained under specific pathogen-free conditions. Animal care was conducted in accordance with the guidelines of Chiba University.

**Cell preparation of Tpath2 cells and ILC2s**. Supplementary Figure 1a and b show the procedure of the preparation of Tpath2 cells and ILC2s. Splenic CD62L[+]KJ1[+]CD4[+] T cells from DO11.10 OVA-specific TCR Tg mice were stimulated with OVA peptides (Loh15, 0.3 μM) plus antigen-presenting cells (irradiated splenocytes) in the presence of IL-2 (25 U mL[−1]), IL-4 (100 U mL[−1]) and anti-IFNγ monoclonal antibody (Th2 cell-skewed condition) for 6 days in vitro. The effector Th2 cells ($3 \times 10^7$) were transferred intravenously into BALB/c *nu/nu* recipient mice. More than 4 weeks after the cell transfer, KJ1[+]CD4[+] T cells in the spleen or lung were purified by autoMACS (Miltenyi Biotec) and cell sorting (BD Aria III) and then used as memory Th2 cells. For the preparation of ILC2, mice were injected with recombinant murine IL-33 (0.5 μg per mouse) intraperitoneally for consecutive 5 days. Tpath2 cells were stained with anti-CD4-APC, anti-KJ1-26-FITC and anti-ST2-Bv421, and purified from memory Th2 mice (Figs. 1a–c, e, f, 2, 3a, b, and 5c). KJ1[+]CD4[+]ST2[hi] memory Th2 cells cultured with IL-7 (10 U mL[−1]), IL-25 (10 ng mL[−1]), and IL-33 (10 ng mL[−1]) were defined as Tpath2 cells. Freshly prepared memory Th2 cells (KJ1[+]CD4[+]) were used for CRISPR/Cas9 experiments (Figs. 3c–h and 6j, k). For purification of ILC2s, spleen cells from memory Th2 mouse (shown in Supplementary Figure 1a) (Figs. 1a–c, e, f, 2, 3a, b, and 5c) or from normal BALB/c mice (Figs. 4b–g, 5d, g, and 6a–i) were stained with anti-lineage cocktail-FITC, anti-CD127-APC, anti- TCRβ-PE, anti-ST2-Bv421 and anti-Thy1.2-PE/Cy7. Lin[−]CD127[+]Thy1.2[+]TCRβ[-] cells cultured with IL-7, IL-25, and IL-33 were defined as ILC2s. Cultured cells were rested with only IL-7 for 24 h. After re-stimulation of Tpath2 cells and ILC2s, each cell population was

stained with anti-IL-5-APC, anti-IL-13-PE, and anti-ST2-Bv421 (Figs. 1a, d, e, 3e, and 4c). Alexa488-conjugated anti-phospho-p38 was used for the staining of Tpath2 cells and ILC2s after stimulation (Figs. 3a, h and 4f). Anti-Human NGFR-Bv421 was used for the isolation of infected ILC2s (Fig. 4 and supplementary Fig. 4). ST2 expression of the infected ILC2s was evaluated by staining with anti-ST2-Bv421 (Supplementary Fig. 4b). Anti-Lineage Cocktail-FITC, anti-CD127-APC, anti-ST2-Bv421 and anti-Thy1.2-PE/Cy7 were used for the detection of ILC2s.

For the preparation of effector Th1 and effector Th17 cells, sorted splenic $CD62L^+CD4^+$ T cells from normal BALB/c mice were stimulated with plate-coated anti-TCRβ (1 μg mL$^{-1}$) plus anti-CD28 antibodies (1 μg mL$^{-1}$) in the presence of IL-2 (25 U mL$^{-1}$), IL-12 (10 U mL$^{-1}$) and anti-IL-4 monoclonal antibody (Th1 cell-skewed conditions) for 6 days in vitro, and in the presence of IL-6 (10 ng mL$^{-1}$), TGFβ (5 ng mL$^{-1}$) and anti-IFNγ and anti-IL-4 monoclonal antibody (Th17 cell-skewed conditions) for 4 days in vitro. ILC1s and ILC3s were purified from the spleen of normal BALB/c mice. Lin$^-$NK1.1$^+$Nkp46$^+$ cells were defined as ILC1s, Lin$^-$CD127$^+$ NK1.1$^-$Nkp46$^{+/-}$ cells and Lin$^-$CD127$^+$CCR6$^+$NK1.1$^-$Nkp46$^-$ cells were defined as ILC3s as described[12].

**Reagents**. The reagents used in this study are as follows: APC- or Bv510-conjugated anti-CD4, FITC- and PE/Cy7-conjugated KJ1-26 (anti-clonotypic for D11.10 TCR), FITC-conjugated anti-CD44, FITC-conjugated anti Lineage Cocktail (containing CD3, Gr-1, CD11b, B220, Ter119), APC- and Bv510-conjugated anti-CD127 (IL-7Rα), APC-conjugated anti-IL-5, PE/Cy7-conjugated anti-Thy1.2, PE-conjugated anti-TCR-beta, PE-conjugated anti-CD335 (Nkp46), Bv421-conjugated ST2, Bv421-conjugated anti-human NGFR and Zombie NIR$^{TM}$ fixable viability kit were purchased from BioLegend, San Diego, CA.

PE-conjugated anti-CCR6 was purchased from R&D systems. Alexa488-conjugated anti-phospho-p38 was purchased from Cell signaling. PE-conjugated anti-IL-13 was purchased from eBioscience. Anti- PE mouse anti-Human NGFR was purchased from BD Biosciences. Anti-IFNγ monoclonal antibody was purchased from BioLegend. Recombinant mouse IL-4 was from TOYOBO, Osaka, Japan. Recombinant mouse IL-33 and mouse IL-25 were purchased from R&D. Recombinant murine IL-7 was purchased from PeproTech. The OVA peptide (residues #323-339; ISQAVHAAHAEINEAGR) was synthesized by BEX Corporation, Tokyo, Japan. SB203580 (Merck) were used for inhibition of p38 activity. Anti-mouse Thy1.2 antibody was purchased from BioX Cell. Detailed information of antibody is listed in Supplementary Table 4.

**Flow cytometry and cell sorting**. Tpath2 cells were stained with anti-CD4-APC, anti-KJ1-26-FITC, and anti-ST2-Bv421, and were purified by FACS ARIA III. ST2$^{hi}$ memory phenotype (MP) CD4 T cells from the spleen of the naïve BALB/c mice were stained with anti-CD4-APC and anti-ST2-BV421 and were purified by FACS ARIA III. ILC2s were stained with anti-Lineage Cocktail-FITC, anti-CD127-APC, anti-Thy1.2-PE/Cy7 and anti-TCR-beta-PE, and were purified by FACS ARIA III in parallel with the isolation of Tpath2 cells. Flow cytometric data were analyzed with Flowjo software (Tree star).

**ELISA for the measurement of cytokine concentration**. Cells were stimulated with PMA (0.1 μg mL$^{-1}$)/Ionomycin (0.5 μM) or IL-7 (2.5 U mL$^{-1}$) /IL-33 (10 ng mL$^{-1}$) in 96-well flat bottom plates for 12 or 48 h at 37 °C. The concentration of IL5 and IL-4 were assessed by ELISA as described previously[23]. The concentration of IL-13 was evaluated with a Mouse IL-13 ELISA Ready-SET-Go! (eBioscience) according to the manufacturer's protocol.

**Quantitative real-time PCR**. Total RNA was isolated with the TRIzol reagent (Invitrogen). cDNA was synthesized with an oligo (dT) primer and Superscript II RT (Invitrogen). Quantitative RT-PCR was performed with the Applied Biosystems StepOnePlus$^{TM}$ Real-Time PCR Systems as described previously[23]. Primers and TaqMan probes were purchased from Applied Biosystems. Primers and Roche Universal probes were purchased from Sigma and Roche, respectively. Gene expression was normalized with the *Hprt* mRNA signal or the *18S* ribosomal RNA signal. Primer sequences are listed in Supplementary Table 1.

**Chromatin immunoprecipitation (ChIP) assay**. ChIP assay was performed as described previously[23]. The antibodies using in ChIP assay as follows; anti-trimethyl histone H3-K4 (Millipore), anti-acetyl histone H3-K9 (Millipore), anti-trimethyl histone H3-K27 (Millipore) and normal rabbit IgG (Santa Cruz). The specific primers used in ChIP assay are described in Supplementary Table 2.

**RNA sequencing**. We isolated ST2$^{hi}$ memory Th2 cells and ILC2s from memory Th2 mice (Supplementary Fig. 1a, and Cell Preparation paragraph in the Methods section). After further cultivation with IL-7, IL-25 plus IL-33, cells were cultured with IL-7 alone for resting for 24 h. Total cellular RNA was extracted with TRIzol reagent (Invitrogen). For cDNA library construction, we used TruSeq RNA Sample Prep Kit v2 (Illumina) according to the manufacturer's protocol. Sequencing the library fragments was performed on an illumina HiSeq 1500 system. For data

analysis, read sequences (50 bp) were aligned to the mm10 mouse reference genome (UCSC, December 2011) using Bowtie (version 0.12.8) and TopHat (version 1.3.2). Fragments per kilobase of exon per million mapped reads (FPKM) for each gene were calculated using Cufflinks (version 2.0.2). Genes with an absolute FPKM of >1 (mean from duplicate samples) were defined as expressed genes. Quality control measures of cDNA libraries were performed on the 2100 Bioanalyzer (Agilent Technologies).

**Cas9-mediated genome editing of *Dusp10***. Four short-guide RNAs were designed using the online tool provided by CHOPCHOP (http://chopchop.cbu.uib.no) or E-CRISP (http://www.e-crisp.org/E-CRISP/). Oligonucleotides pairs with BbsI-compatible overhangs were annealed and cloned into the expression vector pGEM-T Easy-T3-BB-sgRNA for in vitro transcription. Memory Th2 cells (bulk targeted cell cultures) were activated with plate-bounded anti-TCRβ and CD28 antibodies. Cas9 proteins were prepared immediately before experiments by incubating 1 μg Cas9 with 0.3 μg sgRNA indicated below in transfection buffer at room temperature for 10 min. Twenty-four hours after T cell activation, these cells were electroporated with a Neon transfection kit and device (Thermo). And then, these cells were cultivated with IL-7, IL-25, and IL-33 for 4 days for further expansion.

sg*Dusp10*: 5'- CTTGAGGGTCACAACGGCGG-3'
sgControl (luciferase): 5'-CGTATTACTGATATTGGTGGG-3'

**Retroviral vectors and infection**. The retrovirus vector, pMXs-IRES-hNGFR, was provided by Dr. Kitamura (The University of Tokyo, Tokyo, Japan). The method used to generate the virus supernatant and for infection were described previously[23]. Infected cells were collected 3–5 days after stimulation and were subjected to an intracellular cytokine staining, quantitative RT-PCR.

**Immunoblotting and immunoprecipitation**. Immunoblotting was performed as described previously[23]. Briefly, cytoplasmic extracts and nuclear extracts were prepared using NE-PER Nuclear and Cytoplasmic Extraction Reagent (Thermo Fisher Scientific). The antibodies used for the immunoblot analysis were anti–Phospho-p38 (Cell signaling), anti-p38 (Cell signaling), anti–PhosphoGATA3 (abcam), anti-GATA3 (HG3-31; Santa Cruz Biotechnology) and anti-Tubulin (NeoMarkers). Immunoprecipitation was performed as described previously[22]. Flag-tagged GATA3 and Myc-tagged DUSP10 WT and mutant were generated by PCR-based mutagenesis. The expression plasmids were transfected into 293T cells (kindly gifted from Dr. Toshio Kitamura, University of Tokyo) using TransIT-LT1 transfection reagent according to the manufacturer's protocol. A mAb to p-38 (Cell signaling), Flag (M2; Sigma-Aldrich) and mAb to Myc (PL14; MBL, Japan) were used for immunoblot analysis. After immunoprecipitation with mAb to Flag or mAb to Myc, the immunoprecipitates were eluted, and were then separated by electrophoresis.

**Pull-down assay**. Flag-tagged GATA3 and Myc-tagged DUSP10 WT were generated by PCR-based mutagenesis. The expression plasmids were transfected into 293T cells using TransIT-LT1 transfection reagent according to the manufacturer's protocol. In some experiments, graded amounts (0, 1, and 3 μg) of pCMV-Myc-*Dusp10* and pFlag-CMV2-*Gata3* (1 μg) was separately transfected into 293T cells. Cell lysates from 293T cells were incubated with biotinylated oligonucleotides. Bound protein was eluted and separated on an SDS polyacrylamide gel and then subjected to immunoblotting with specific antibodies. Double-stranded oligonucleotide probes for the pull-down assay are described at Supplementary Table 3.

**Luciferase reporter assay**. A single copy of an *Il5* promoter in the luciferase reporter plasmid, pGL3 Basic (Promega) and a renilla luciferase plasmid (qRL; Promega), were used. D10G4.1 cells were used for transfection by electroporation. Twenty-four hours after electroporation, cells were stimulated with PMA plus dibutyryl cyclic AMP overnight. The cell extracts were prepared and subjected to a luciferase assay using the manufacturer's instructions for the Dual-luciferase reporter (Promega).

**In vivo experiment**. Thy1.1$^+$ILC2s were infected with a *Dusp10*- or Mock- IRES-hNGFR-containing retrovirus. Infected cells were cultured with IL-7 (25 U mL$^{-1}$), IL-25 (10 ng mL$^{-1}$), and IL-33 (10 ng mL$^{-1}$) in vitro for 4 days and purified by cell sorting (BD Aria III) after cultivation. $1 \times 10^5$ cells were transferred intravenously into Thy1.2$^+$ BALB/c *nu/nu* mice. Mice were injected intraperitoneally with Thy1.2 antibody on days 0, 1, and 3, and treated with intranasal injection of papain (50 μg) or recombinant mouse IL-33 (rmIL-33: 10 ng) in 20 μL in sterile PBS on days 1, 2, and 3. On day 4, bronchoalveolar lavage fluid (BALF) was collected at 24 h after from last challenge with papain or rmIL-33 and lung histology was assessed. NSG mice that received with Mock (empty vector)- or *Dusp10*-transduced ILC2s (Fig. 6g–i), or with control- or sg*Dusp10*-memory Th2 cells (Fig. 6j, k) were treated with intranasal injection of rmIL-33 on days 1, 2, and 3. Airway hyperreactivity was assessed by methacholine-induced (Sigma-Aldrich) airflow obstruction at 24 h after

the last IL-33 administration using a computer-controlled small animal ventilator (SCIREQ)[23]

**Statistical analysis and general methods.** Data are expressed as mean ± SD or mean ± SEM. The data were analyzed with the Graphpad Prism software program (version 6). Differences were assessed using two-tailed Student t tests, or non-parametric Mann–Whitney $U$-test where appropriate. Differences with $p$ values of <0.05 were considered to be significant. Sample size for animal studies was chosen based on prior experience with similar models of memory cell formation. No data were excluded from the analysis of experiments. Mice were commercially sourced and randomized into experimental groups upon arrival, and all animals within a single experiment were processed at the same time. For cell sorting and RNA-seq analysis the investigator was blinded. For cell transfer experiments the investigator was not blinded. Data display similar variance between groups and are normally distributed where parametric tests are used.

## Data availability

RNA-sequence data for Tpath2 cells, ILC2s, and Th2 cells are available in the Gene Expression Omnibus (GEO) database (http://www.ncbi.nlm.nih.gov/geo) under accession number GSE116842.

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

## Acknowledgements

We thank Kaoru Sugaya, Miki Kato, and Toshihiro Ito for their excellent technical assistance. This work was supported by Japan Science and Technology Agency (JST), CREST, and by grants from the Ministry of Education, Culture, Sports, Science and Technology (MEXT Japan) (Grants-in-Aid: for Scientific Research [S] #26221305, [B] #21390147, and [C] #15K06838, #17K08876, #18K07257, Young Scientists [A] #16H06224, and [B] #24790461, Challenging Exploratory Research #23659240, Scientific Research on Innovative Areas (Research in a proposed research area) 18H04665, AMED-CREST, AMED (no. JP16gm0410009); Practical Research Project for Allergic Diseases and Immunology (Research on Allergic Diseases and Immunology) fron AMED (no. JP18ek0410030). The Astellas Foundation for Research on Metabolic Disorders, The Uehara Memorial Foundation, Osaka Foundation for Promotion of Fundamental Medical Research, Kanae Foundation for the Promotion of Medical Science, Ono Medical Research Foundation, and Takeda Science Foundation.

## Author contributions

T.Y., Y.E., D.J.T., N.S., and To.N. conceived and directed the project, designed experiments, interpreted the results, and wrote the manuscript. T.Y. and Y.E. designed the project, analyzed main experiments. T.Y., Y.E., A.O., K.H., Ta.N., T.K., and H.K.A. developed experimental protocols and performed experiments. Y.O. and S.U. designed and performed CRISPR/Cas9 experiment. H.N. and O.N. prepared the Cas9 protein.

## Additional information

**Competing interests:** The authors declare no competing interests.

All data that support the findings of this study are available from the corresponding author upon reasonable request.

