## [Peer Review File · Nature Communications]

Reviewers' comments:

Reviewer #1 (IL33, IL1 family cytokine)(Remarks to the Author):

In this study, Yamamoto et al. investigate the difference in the regulation of cytokine production between type 2 innate lymphoid cells (ILC2) and a particular type of T helper (Th)2 cells, that they call Tpath2 cells. They report that stimulation with cytokines, such as IL-33, alone induces IL-5 and IL-13 production in ILC2, but not in Tpath2 cells. The authors further investigate the molecular mechanisms underlying this fundamental difference and identify a MAPK phosphatase, DUSP10, that is differentially expressed in the two cell types. The authors finally describe that DUSP10 acts as a negative regulator of IL-33-induced cytokine production in Tpath2 cells by preventing the phosphorylation and activation of GATA3 by p38 MAPK.

The topic of this study is very interesting and original. However, as detailed below in my comments to the authors, the description of the methods and experiments performed is sometimes sketchy and additional information needs to be provided. In addition, some conclusions seem overstated and insufficiently supported by data. Finally, throughout the manuscript, a number of qualitative data are used to convey information concerning quantitative differences between samples. In these cases, quantitative analysis of the data, with statistics, needs to be provided also.

Specific comments:

1. According to the explanatory scheme provided in Supplementary Figure 1a-b, ILC2s were isolated either from naïve BALB/c mice or from what the authors call 'memory Th2' mice. They further describe in 'materials and methods' that ILC2s were isolated from spleen or lung. It is not clear to which extent these alternative sources for ILC2 isolation affect the results. For instance, IL-5 production looks different in ILC2s in Figure 1a and 1d. It should be more clearly described for each experiment/Figure what source of ILC2 was used.

Supplementary Figure 1a is not completely clear for me. Please define explicitly what are the 'memory Th2' mice used for ILC2 isolation. Are these also BALB/c-nu/nu mice transferred with effector Th2 cells, as for Tpath2 isolation?

2. Materials and methods, flow cytometry and cell sorting: the protocol provided describes only the sorting, but not the stainings shown in Figure 1 (IL-5, IL-13, ST2), Figure 3 (phospho-p38, IL-5, IL-13), Figure 4 (hNGFR, ST2) and Suppl. Figure 5 (Thy1.2). Protocols, or at least references, need to be provided. Also the gating strategies used for sorting and for the different multicolor stainings need to be shown.

3. Materials and methods, in vivo: the methods for the experiments shown in Fig 6h, i and j are missing.

4. Figure 2:

In Figure 2a: IL-5 and IL-13 show up in the group of highly expressed genes that are common to both Tpath2 and ILC2, rather than in a group of genes differentially expressed between these two cell types. This seems at odds with the data from Figure 1 and with the general message of the manuscript. Similarly, Dusp2 shows up as differentially expressed in Figure 2a and c, but its (low) expression levels look similar in d. The authors should comment on these discrepancies between RNAseq and RTqPCR data.

5. Figure 3:

Figure 3a: The differences in the MFI shift between IL7/IL33 stimulated Tpath2 and the other panels is

not very striking. How reproducible/relevant is this difference? In order to validate this observation, quantitative data (with statistical analysis) for the n=3 independent experiments need to be provided. The same holds true for Figure 3h.

Figure 3c: the methods for the CRISPR/Cas9 targeting of Tpatho2 need to be described in more detail. In particular, it is not clear whether bulk targeted cell cultures or isolated single cell clones were used for the different in vitro and in vivo experiments. Also, what exactly are the 'Cas9 control' cells used? Please provide protocol details.

The text (p.11, line 228) states that editing was performed in primary effector T cells, while Supplementary Figure 3a and the 'materials and methods' section mention the use of freshly isolated memory Th2 cells activated with anti-TCR and anti-CD28 antibodies. Please clarify. In any case, these cells are different from the Tpatho2 cells used in the other experiments, so please explain how these data are relevant and informative with regard to the situation in Tpatho2 cells.

6. Figure 4: Data confirming efficient overexpression of the DUSP10 WT or mutant protein (or at the very least mRNA) are missing from this Figure and need to be provided.

Figure 4e and h show again qualitative data, which are meant to convey information about quantitative differences. In order to validate these observations, quantitative data for the n>4 independent experiments need to be provided.

Figure 4h, data obtained with the mutant DUSP10 C409S variant would be informative as well.

Figure 4e,f,g show only stimulated cells. Data for baseline cytokine production in resting cells should be shown also.

7. Figure 5:

Figure 5b: phospho-p38 is detected in lanes 1 and 2 (untransfected and GATA3 only transfected cells), however in Supplementary Figure S4, there is no phospho-p38 in untransfected cells, neither in cytosol (although there is total p38, lane 1), nor in nucleus (lanes 5), so where is it? How can these contradictory findings go together?

Text, p.15, line 301-303: 'these results indicate that DUSP10 suppresses phosphorylation of GATA3, and thereby its nuclear translocation.' this last statement is not correct, since the same amount of GATA3/Flag is detected in the nuclear fraction with or without DUSP10 cotransfection (Supplementary Figure 4). Please amend the text.

Figure 5d: a condition showing unstimulated cells with baseline GATA3 phosphorylation levels is missing to demonstrate that phosphorylation is indeed induced by IL7/IL33.

Figure 5e: more details on the pull-down procedure need to be provided. How were the gradients obtained? Cotransfection of increasing amounts DNA? What do the numbers 0,1 and 3 in the Myc-DUSP10 line refer to? Please clarify. The experiment could be better validated by addition of a mutant oligonucleotide as a negative control.

Figure 5f: what does the 'GATA3' label on the x-axis refer to? Cotransfection of a GATA3 expression plasmid? Please complete methods and figure legend.

8. Figure 6a-g: as for Figure 4, efficient overexpression of the DUSP10 protein in the transduced ILC2 needs to be documented.

Figure 6h, similarly, additional information on efficient DUSP10 targeting in Thpatho2 cells needs to be provided. Was lack of DUSP10 expression verified before injection and how?

The authors' conclusion (p.19, lines 385-387) that 'DUSP constrains...by modulating cytokine production by ILC2' is not supported by any data here. Cytokine levels in BALF need to be shown, or the conclusion amended.

Minor comments:

1. Throughout the paper, the authors talk about the effect of DUSP10 expression on IL-33-induced cytokine production, when, in fact (except for Fig.1a) what was investigated was cytokine production

induced by IL-33 in combination with IL-7, and not by IL-33 alone. The text of the manuscript should be amended accordingly.

2. Materials and methods, mice: the authors state that 'animals used in this study were backcrossed 10 times to BALB/c'. Surely this does not apply to all mouse lines (e.g. C57BL/6, NSG..). What about the Thy1.1 mice? Please specify and provide complete systematic names for all mouse strains.

3. Supplemental methods, reagents: the text includes an anti-human CD271 antibody, which has not been used in this study.

4. Supplemental methods, RNA sequencing: information on quality control and bioinformatics analyses for sample comparison is missing.

5. Supplemental methods, pull-down assay: expression plasmids need to be described in more detail: backbone, GenBank accession numbers for the cDNA sequences used...
The oligonucleotide used for the pull-down was presumably double-stranded? Please specify.

6. Figure 1f: please provide a protocol or reference for Th1 cell generation.

7. Figure 2a: were the Tpath2 and ILC2 cells which were used for the RNAseq experiment resting or stimulated with cytokines? Please specify.

8. Figure 4: MOCK presumably stands for 'empty vector'? Please clarify.

9. Supplementary Figure 1d: there is no U4 region in the Figure to go with the U4 sequence provided in the legends.

Reviewer #2 (IL33, ST2, inflammation)(Remarks to the Author):

This is a very well constructed and presented report full of cutting-edged technology and convincing data. My only question is whether it has a major impact in the field of Immunology in general and in inflammation in particular to claim the space for NC.

The role of DUSPs in reduction of inflammation is known (eg. Ref. 44). DUSP^{-/-} macrophages have enhanced inflammatory cytokines production upon TLR stimulation has also been reported (eg. Ref. 46). The major contribution of the present report is that there is a distinct difference between memory Th2 and ILC2 (both express IL-33R) in their response to IL-33. Furthermore, the difference is explained by the high expression of DUSP10 in Th2 cells but not in ILC2. This is fair enough. However, what is the implication of this in term of the overall immune response (eg. to allergy) is unclear. In another words, what is the evolutionary advantage of such a difference? The second question is whether this is a general phenomenon or rather restricted to Th2. Thirdly, whether this also applies to humans is also unknown.

Reviewer #3 (Airway inflammation, Th2)(Remarks to the Author):

The authors provide a follow up on their previous finding of ST2⁺ allergen-specific memory-type pathogenic Th2 (T_{path2}) cells in allergic eosinophilic airway inflammation. In this study, the authors identify DUSP10, a p38/MAPK/JNK phosphatase, as a key negative regulator of IL-33-induced cytokine production in T_{path2} cells through p38-GATA3 axis to explain the fundamental difference between ILC2 and TH2 cells and why T_{path2} cells do not produce IL-5 and IL-13 in response to IL-33 like ILC2s. This finding is consistent with the current understanding of DUSP10 as a negative regulator of effector T cell (both TH1 and TH2) cytokine expression¹. As outlined below, the manuscript requires clarification of several issues.

General. All flow cytometry analyses include only representative figures and lack. This is not acceptable- fully quantified data including statistical analyses, number of replicates, etc. must be shown.

Figure 1. Parts b and c: the fold increase in IL-13 gene and protein expression induced by PMA and Ionomycin (P+I) as compared to IL-7 and IL-33 stimulation in T_{path2} cells is significantly lower than that in flow cytometry analysis in part a, realizing that the quantification method between these techniques is not the same. Please clarify this and include the quantification plot for flow. Part d: The authors have mentioned that ST2^{hi} MPCD4 T cells do not produce IL-5 and IL-13 in response to IL-7 plus IL-33 as ILC2 does, however, the flow analysis also shows that the MPCD4 T cells do not respond much to P+I, the positive control. This makes the claim questionable as the cells do not seem to respond well under any circumstances.

Figure 3. Part b: in the figure legend, the authors mention total p38 expression, but these data are not shown in the actual figure. In addition, the intensities of the endogenous control, Tub. α , in blots between ILC2 and T_{path2} cells vary significantly, impeding direct comparison of p-p38 expression between the two types of cells when quantification plots are lacking-please perform densitometry to assess the ratio of p-p38 and tubulin alpha to provide more accurate relative quantification. The authors utilize CRISPR-Cas9 system to delete Dusp10 in primary effector cells to study the role of it in IL-33-mediated cytokine production in T_{path2} cells. However, DUSP10 (MKP5) knock-out mice are available and have been used in several studies in the past^{1, 2, 3, 4}. Is there a reason why CRISPR-Cas9 is preferential to the knock-out mice? It seems that in subsequent experiments, it is more straightforward to using the KO mice, especially in in vivo studies.

Figure 4. Part h: the authors claim that overexpression of DUSP10 results in decreased expression of p-p38 in ILC2s compared to the mock-infected ILC2s in response to IL-7 and IL-33 stimulation. However, while the MFI of p-p38 is indeed lower, the shift from the flow histograms between the two experimental groups is very subtle and unconvincing. Again, please provide quantification plots with statistics and also demonstrate the observation through western blots to compliment the flow analysis.

Figure 5. The authors nicely show the physical interaction among DUSP10, p38 and GATA3 and its dependency on IL-33. It would strengthen the conclusion and add mechanistic insights if the authors can show the role of IL-33 receptor, ST2, in this by using blocking antibody or inhibitor.

Figure 6. As mentioned previously, it would further complement the results to use MKP5 KO mice in addition. While the authors examine cellular infiltration to BALf as well as inflammation and mucus production by H&E and PAS staining to evaluate allergic inflammation in both papain and IL-33-induced airway inflammation models, it would strengthen this figure if additional allergic inflammatory parameters are examined as well, such as airway hyperresponsiveness, TH2 cytokine production in the lung and expression of ST2, MUC5AC and GATA3 etc.

Overall significance. The findings, if confirmed, are interesting biochemically, but it remains unclear at a practical level what the significance of the findings are. For example, it has now been shown definitively that IL-33 is not important to allergic inflammation and disease in diverse contexts^{5, 6}. Moreover, as the authors note, the importance of ILCs in general to human immunity has now been questioned⁷. Together, these findings seem to challenge the conclusion in the manuscript that, "...DUSP10 may play a critical role in the control of IL-33-associated allergic airway inflammation both

[in] mouse and human systems". Ideally, the authors would take on these challenges and attempt to place their findings in this more complete context of the regulation of the type 2 immune response. A more convincing story would further be possible if the authors examined DUSP10 as a potential therapeutic to allergic asthma by examining whether adding DUSP10 or adaptively transferring DUSP10-overexpressed ILC2s and TH2 cells rescued mice from developing allergic airway inflammation. Since GATA3 is incontrovertibly critical for TH2 cell differentiation (and suppressing TH1 cell differentiation), in addition to promoting TH2 cytokine production, the authors could further investigate whether IL-33-DUSP10-p38 axis has a role in TH1 and TH2 cell differentiation.

References

1. Zhang, Y. et al. Regulation of innate and adaptive immune responses by MAP kinase phosphatase 5. *Nature* 430, 793-797 (2004).
2. Qian, F. et al. Map kinase phosphatase 5 protects against sepsis-induced acute lung injury. *American journal of physiology. Lung cellular and molecular physiology* 302, L866-874 (2012).
3. Cheng, Q. et al. MAPK phosphatase 5 deficiency contributes to protection against blood-stage *Plasmodium yoelii* 17XL infection in mice. *Journal of immunology* 192, 3686-3696 (2014).
4. Shi, H. et al. Improved regenerative myogenesis and muscular dystrophy in mice lacking Mkp5. *The Journal of clinical investigation* 123, 2064-2077 (2013).
5. Hoshino, K. et al. The absence of interleukin 1 receptor-related T1/ST2 does not affect T helper cell type 2 development and its effector function. *J Exp Med* 190, 1541-1548 (1999).
6. Vannella, K.M. et al. Combinatorial targeting of TSLP, IL-25, and IL-33 in type 2 cytokine-driven inflammation and fibrosis. *Sci. Transl. Med.* 8, 337ra365 (2016).
7. Vely, F. et al. Evidence of innate lymphoid cell redundancy in humans. *Nat Immunol* 17, 1291-1299 (2016).

David B. Corry, M.D.

Reviewer #1 (IL33, IL1 family cytokine)(Remarks to the Author):

In this study, Yamamoto et al. investigate the difference in the regulation of cytokine production between type 2 innate lymphoid cells (ILC2) and a particular type of T helper (Th) 2 cells, that they call Tpath2 cells. They report that stimulation with cytokines, such as IL-33, alone induces IL-5 and IL-13 production in ILC2, but not in Tpath2 cells. The authors further investigate the molecular mechanisms underlying this fundamental difference and identify a MAPK phosphatase, DUSP10, that is differentially expressed in the two cell types. The authors finally describe that DUSP10 acts as a negative regulator of IL-33-induced cytokine production in Tpath2 cells by preventing the phosphorylation and activation of GATA3 by p38 MAPK.

The topic of this study is very interesting and original. However, as detailed below in my comments to the authors, the description of the methods and experiments performed is sometimes sketchy and additional information needs to be provided. In addition, some conclusions seem overstated and insufficiently supported by data. Finally, throughout the manuscript, a number of qualitative data are used to convey information concerning quantitative differences between samples. In these cases, quantitative analysis of the data, with statistics, needs to be provided also.

Thank you for this reviewer's critical reading and suggestions to improve our manuscript.

Specific comments:

1. According to the explanatory scheme provided in Supplementary Figure 1a-b, ILC2s were isolated either from naïve BALB/c mice or from what the authors call 'memory Th2' mice. They further describe in 'materials and methods' that ILC2s were isolated from spleen or lung. It is not clear to which extent these alternative sources for ILC2 isolation affect the results. For instance, IL-5 production looks different in ILC2s in Figure 1a and 1d. It should be more clearly described for each experiment/Figure what source of ILC2 was used. Supplementary Figure 1a is not completely clear for me. Please define explicitly what are the 'memory Th2' mice used for ILC2 isolation. Are these also BALB/c-nu/nu mice transferred with effector Th2 cells, as for Tpath2 isolation?

Response:

As suggested, we clarified the source for ILC2 isolation for each experiment/Figure in the Figure legend and Materials and Methods section in revised manuscript. Furthermore, we modified the experimental scheme in **Supplementary Figure 1a and 1b**. In addition, we re-wrote the procedure for the generation of memory Th2 mice and the preparation of Tpath2 cells and ILC2s in the paragraph of **Cell preparation of Tpath2 cells and ILC2s** in the Materials and Methods section in more detail. We included these in the revised manuscript.

2. Materials and methods, flow cytometry and cell sorting: the protocol provided describes only the sorting, but not the stainings shown in Figure 1 (IL-5, IL-13, ST2), Figure 3 (phospho-p38, IL-5, IL-13), Figure 4 (hNGFR, ST2) and Suppl. Figure 5 (Thy1.2). Protocols, or at least references, need to be provided. Also the gating strategies used for sorting and for the different multicolor stainings need to be shown.

Response:

As suggested, we described the protocols for our staining and gating strategies in the **Cell preparation of Tpath2 cells and ILC2s** paragraph in the Materials and Methods section.

3. Materials and methods, in vivo: the methods for the experiments shown in Fig 6h, i and j are missing.

Response:

As suggested, we described the methods for the experiments shown in new **Fig. 6j and k** and **Supplementary Fig. 6g** and in the Materials and Methods section in the revised manuscript.

4. Figure 2:

In Figure 2a: Il-5 and Il-13 show up in the group of highly expressed genes that are common to both Tpath2 and ILC2, rather than in a group of genes differentially expressed between these two cell types. This seems at odds with the data from Figure 1 and with the general message of the manuscript. Similarly, Dusp2 shows up as differentially expressed in Figure 2a and c, but its (low) expression levels look similar in d. The authors should comment on these discrepancies between RNAseq and RTqPCR data.

Response:

The central message of the manuscript is that IL-33 cannot directly induce IL-5 and IL-13 from ST2⁺ Tpath2 cells (without stimulation from PMA + Ionomycin

or TCR signaling), but can from ST2⁺ ILC2, and that this difference is mediated by *Dusp10*. We have modified the groups included in the analysis in **Fig. 2** in order to avoid any misunderstandings. The newly generated venn diagram is shown in **Fig. 2a**. In the previous manuscript, we included the results of both unstimulated and stimulated (PMA + Ionomycin or IL-7 + 33) samples in the venn diagram. Thus, several cytokines strongly induced in response to PMA + Ionomycin stimulation in Tpath2 cells were included. As displayed in the new Figure legend of **Fig. 2a**, we now show only the results of “unstimulated” -Tpath2 cells and -ILC2s. These groups are better suited to examine the central phenomenon described in the manuscript. Cytokines including *Il5*, *Il13*, *Areg* and *Il6* that are elevated by stimulation are not shown in the revised figure. We revised the legend for **Fig. 2a** “Immune response-related genes (GO0006955) preferentially expressed by unstimulated Tpath2 cells (Red) or unstimulated ILC2s (Blue) (> 5-fold difference), or genes shared by the two (Purple) (< 1.2-fold difference) are shown. Transcripts expressed above 10 fragments per kilobase of exon per million reads mapped (FPKM) are shown.”

Regarding the discrepancy in the expression of *Dusp2*, although the levels of *Dusp2* appeared to be differentially expressed in RNA-sequencing analysis, the magnitude of its gene expression was very low in qRT-PCR analysis (**Fig. 2c and d**). We included a comment regarding the discrepancies in these figures in the revised manuscript (p.11).

5. Figure 3:

Figure 3a: The differences in the MFI shift between IL7/IL33 stimulated Tpath2 and the other panels is not very striking. How reproducible/relevant is this difference? In order to validate this observation, quantitative data (with statistical analysis) for the n=3 independent experiments need to be provided. The same holds true for Figure 3h. Figure 3c: the methods for the CRISPR/Cas9 targeting of Tpatho2 need to be described in more detail. In particular, it is not clear whether bulk targeted cell cultures or isolated single cell clones were used for the different in vitro and in vivo experiments. Also, what exactly are the ‘Cas9 control’ cells used? Please provide protocol details. The text (p.11, line 228) states that editing was performed in primary effector T cells, while Supplementary Figure 3a and the ‘materials and methods’ section mention the use of freshly isolated memory Th2 cells activated with anti-TCR and anti-CD28 antibodies. Please clarify. In any case, these cells are different from the Tpatho2 cells used in the other experiments, so please explain how these data are relevant and informative with regard to the situation in Tpatho2 cells.

Response:

Regarding the comments for Figure 3a and 3h, we included quantitative data for four independent experiments (*p<0.05). In response to the comments regarding the method used in the CRISPR/Cas9 system, we described the experimental procedure in more detail in the Figure legend and Materials and Methods

sections. “Memory Th2 cells (bulk targeted cell cultures) were activated with plate-bound anti-TCR β and CD28 antibodies. Cas9 proteins were prepared immediately before experiments by incubating 1 μ g Cas9 with 0.3 μ g sgRNA in transfection buffer at room temperature for 10 min. 24 hours after T cell activation, these cells were electroporated with a Neon transfection kit and device (Thermo). And then, these cells were cultivated with IL-7, IL-25, and IL-33 for 4 days for further expansion.” For the description in the text (page 12), we made the sample name consistent (memory Th2 cells) in order to avoid misunderstanding. While freshly isolated memory Th2 cells contain ST2^{hi} and ST2^{lo} populations, the majority (more than 80%) become ST2⁺ after culturing with IL-7, IL-25, and IL-33, almost equivalent levels to Tpath2 cells. The expression of *Dusp10* in memory Th2 cells cultured with IL-7, IL-25, and IL-33 was also comparable to the level detected in Tpath2 cells. Therefore, we think that the data obtained from freshly isolated memory Th2 cells is relevant and informative with regard to the function of *Dusp10* in Tpath2 cells. We included a series of quantitative data, the method in detail used in the CRISPR/Cas9 system, and the description of Th2 cells (memory Th2 cells) in the Result section, Material and Methods section and Figure legend section in the revised manuscript.

6. Figure 4: Data confirming efficient overexpression of the DUSP10 WT or mutant protein (or at the very least mRNA) are missing from this Figure and need to be provided. Figure 4e and h show again qualitative data, which are meant to convey information about quantitative differences. In order to validate these observations, quantitative data for the n>4 independent experiments need to be provided.

Figure 4h, data obtained with the mutant DUSP10 C409S variant would be informative as well. Figure 4e,f,g show only stimulated cells. Data for baseline cytokine production in resting cells should be shown also.

Response:

As suggested, we included the data of the DUSP10 protein levels in the transduced ILC2. We easily detected DUSP10 protein in the transduced ILC2 by western blotting (**Fig. 4b**).

In response to the reviewer’s comment regarding qualitative differences, we included quantitative data for three and four independent experiments (**Fig. 4c and f; *p<0.05**).

We have assessed the effects of the DUSP10 C409S mutant on the p38 phosphorylation, and no obvious inhibitory effect on p38 phosphorylation was observed (see **Figure 4f**).

We included the results for baseline cytokine production in resting cells in the revised manuscript (see **Fig. 4d and e**).

7. *Figure 5:*

Figure 5b: phospho-p38 is detected in lanes 1 and 2 (untransfected and GATA3 only transfected cells), however in Supplementary Figure S4, there is no phospho-p38 in untransfected cells, neither in cytosol (although there is total p38, lane 1), nor in nucleus (lanes 5), so where is it? How can these contradictory findings go together?

Text, p.15, line 301-303: 'these results indicate that DUSP10 suppresses phosphorylation of GATA3, and thereby its nuclear translocation..' this last statement is not correct, since the same amount of GATA3/Flag is detected in the nuclear fraction with or without DUSP10 cotransfection (Supplementary Figure 4). Please amend the text. Figure 5d: a condition showing unstimulated cells with baseline GATA3 phosphorylation levels is missing to demonstrate that phosphorylation is indeed induced by IL7/IL33. Figure 5e: more details on the pull-down procedure need to be provided. How were the gradients obtained? Cotransfection of increasing amounts DNA? What do the numbers 0,1 and 3 in the Myc-DUSP10 line refer to? Please clarify. The experiment could be better validated by addition of a mutant oligonucleotide as a negative control. Figure 5f: what does the 'GATA3' label on the x-axis refer to? Cotransfection of a GATA3 expression plasmid? Please complete methods and figure legend.

Response:

The contradictory results between **Fig. 5b** and **Supplementary Fig. 4** may be due to differences in the procedures of purifying proteins. Specifically, the composition of the lysis buffer was different between these two figures (phosphatase inhibitor was not included in the lysis buffer used in **Supplementary Fig. 4**). We have removed original **Supplementary Fig. 4** from the revised manuscript because the phosphatase inhibitor was not included in this experiment.

As suggested, regarding the nuclear localization of GATA3, we omitted the text describing the role of DUSP10 in GATA3 nuclear localization "*and thereby its nuclear translocation*". We included the baseline levels of GATA3 phosphorylation in **Fig. 5d**. We observed the phospho-GATA3 was increased in IL-7/IL-33 stimulation, and was decreased by the transduction of *Dusp10* (**Fig. 5d**). Furthermore, we included the description of experimental procedures for DNA pull-down experiments in Results section and Materials and Methods section. For gradients of *Dusp10* expression, graded amounts (0, 1, and 3 μg) of pCMV-Myc-*Dusp10* and pFlag-CMV2-*Gata3* (1 μg) was separately transfected into 293 T cells, and DNA pull-down assay was performed as described in the

Material and methods section. The x-axis of GATA3 refers to the amounts of cell lysates (3-fold doses; DNA-precipitants and input samples) that were blotted with anti-Flag Ab. To address the point regarding mutant oligonucleotides, we performed the DNA pull-down assays by adding a mutant oligonucleotide (a mutant //5p oligonucleotide that GATA3 cannot bind) as a negative control (**Supplementary Fig. 5**). We included these series of results in the revised manuscript.

8. Figure 6a-g: as for Figure 4, efficient overexpression of the DUSP10 protein in the transduced ILC2 needs to be documented. Figure 6h, similarly, additional information on efficient DUSP10 targeting in Tpatho2 cells needs to be provided. Was lack of DUSP10 expression verified before injection and how? The authors' conclusion (p.19, lines 385-387) that 'DUSP constrains...by modulating cytokine production by ILC2' is not supported by any data here. Cytokine levels in BALF need to be shown, or the conclusion amended.

Response:

As suggested, we confirmed the DUSP10 protein levels in the transduced ILC2 and targeted Tpath2 cells by Western blotting (**Fig. 3d and 4b**). In response to the reviewer's comments regarding the cytokine levels in BALF, we examined the levels of cytokines by an ELISA, and significantly decreased IL-5 and IL-13 production in the BAL fluid in *Dusp10*-ILC2 group was detected (**Fig. 6h**). We included these results in the revised manuscript.

Minor comments:

1. Throughout the paper, the authors talk about the effect of DUSP10 expression on IL-33-induced cytokine production, when, in fact (except for Fig.1a) what was investigated was cytokine production induced by IL-33 in combination with IL-7, and not by IL-33 alone. The text of the manuscript should be amended accordingly.

Response:

IL-7 was added to the in vitro culture to support cell survival, and does not induce p38-MAPK signaling. Thus, we believe that the effect shown in the manuscript is mediated mainly by IL-33. However, to describe the results more stringently, we used "IL-7/33 stimulation" when we described the experimental results.

2. Materials and methods, mice: the authors state that 'animals used in this study were backcrossed 10 times to BALB/c'. Surely this does not apply to all

mouse lines (e.g. C57BL/6, NSG..). What about the Thy1.1 mice? Please specify and provide complete systematic names for all mouse strains.

Response:

As suggested, we specified the mice which are backcrossed 10 times to BALB/c. The background of Thy1.1 mice (BALB/c) is described in the revised Materials and Methods section.

3. Supplemental methods, reagents: the text includes an anti-human CD271 antibody, which has not been used in this study.

Response:

Anti-human CD271 refers to anti-human NGFR. We made the anti-human NGFR terminology consistent in the supplemental Methods section.

4. Supplemental methods, RNA sequencing: information on quality control and bioinformatics analyses for sample comparison is missing.

Response:

As suggested, we included information on quality control and bioinformatics analyses for the sample comparisons in the supplemental Methods section.

5. Supplemental methods, pull-down assay: expression plasmids need to be described in more detail: backbone, GenBank accession numbers for the cDNA sequences used...

The oligonucleotide used for the pull-down was presumably double-stranded? Please specify.

Response:

As suggested, we included information on the expression plasmids (pFlag-CMV2-*Gata3*, pCMV-Myc-*Dusp10*, and *Dusp10C409S*) and specify the double-stranded oligonucleotides (Wt 115p and Mut115p) used in the pull-down experiments in detail in the supplemental Method section in the revised manuscript.

6. Figure 1f: please provide a protocol or reference for Th1 cell generation.

Response:

As suggested, we included the protocol for Th1 cell differentiation in the Materials and Methods section.

7. *Figure 2a: were the Tpath2 and ILC2 cells which were used for the RNAseq experiment resting or stimulated with cytokines? Please specify.*

Response:

As suggested, we described the cell conditions used for the RNA-seq analysis in the legend for Figure 2.

8. *Figure 4: MOCK presumably stands for 'empty vector'? Please clarify.*

Response:

As the reviewer assumes, Mock means 'empty vector'. We clarified this in the text in order to avoid any misunderstandings.

9. *Supplementary Figure 1d: there is no U4 region in the Figure to go with the U4 sequence provided in the legends.*

Response:

We omitted the U4 region in Supplementary Figure 1d.

Reviewer #2 (IL33, ST2, inflammation)(Remarks to the Author):

This is a very well constructed and presented report full of cutting-edged technology and convincing data. My only question is whether it has a major impact in the field of Immunology in general and in inflammation in particular to claim the space for NC.

The role of DUSPs in reduction of inflammation is known (eg. Ref. 44). DUSP-/- macrophages have enhanced inflammatory cytokines production upon TLR stimulation has also been reported (eg. Ref. 46). The major contribution of the present report is that there is a distinct difference between memory Th2 and ILC2 (both express IL-33R) in their response to IL-33. Furthermore, the difference is explained by the high expression of DUSP10 in Th2 cells but not in ILC2. This is fair enough. However, what is the implication of this in term of the overall immune response (eg. to allergy) is unclear. In another words, what is the evolutionary advantage of such a difference? The second question is whether this is a general phenomenon or rather restricted to Th2. Thirdly, whether this also applies to humans is also unknown.

Response:

To address the first point, we discussed the role of DUSP10 in the overall immune response, such as to allergy and also the evolutionary advantages in the Discussion section. First, by referring a recent report with meta-genome analyses of Th2 cells from the patients of chronic allergic disorders including asthma and rhinitis, the pathophysiological significance of the expression of DUSP10 in Th2 cells in the context of the induction and/or the pathogenesis of allergic diseases are discussed. In our report, we demonstrate a mechanism (differential expression of *Dusp10*) that differentiates the innate function such as cytokine-induced cytokine expression characteristic to ILC2s from antigen-dependent adaptive Th2 cell function. A high level expression of *Dusp10* in T_{path2} cells strictly restrained the innate immune response induced by IL-7/33 stimulation. A possible advantage of the negative regulation of the reactivity to epithelial cell-derived cytokine IL-33 in Th2 cells in the mucosal barrier such as the airway in vertebrates is discussed.

We included two paragraphs to address these issues in the revised manuscript (pages 25 and 26) as follows;

“The results from our mouse airway inflammation models indicate that DUSP10 functions as a negative regulator of IL-33-mediated allergic airway inflammation by limiting Th2 cytokine production in T_{path2} cells in response to IL-33. Recently, global transcriptional profiling to determine pathogenic features associated with asthma and rhinitis has been reported by analyzing Th2 cells from subjects with allergic asthma and rhinitis (Seumois et al. 2016). They found significantly lower levels of *DUSP10* in Th2 cells from asthmatic subjects. Thus, it is conceivable that the decreased expression of *DUSP10* in Th2 cells causes allergic diseases and/or is associated with the pathogenesis of chronic allergic disorders.”

“Interestingly, both ILC1s and ILC3s express high levels of *Dusp10* as compared to ILC2s. However, ILC1s and ILC3s do not express the IL-33 receptor ST2 (Sonnenberg et al. 2015). Reactivity to IL-33 in ILCs is therefore controlled by at least two distinct mechanisms; ST2 expression and *Dusp10* expression. Conventional effector Th2 cells express low levels of *Dusp10*, and limited amounts of ST2 (Endo et al. 2015), while T_{path2} cells express high levels of *Dusp10* and ST2. Thus, T_{path2} cells may acquire increased expression of ST2 and also high expression of *Dusp10* during their differentiation from effector Th2 cells. In any event, reactivity to IL-33 in T_{path2} cells appears to be strictly regulated, particularly in the IL-33-abundant environment of the inflamed airway (Drake et al. 2017). Indeed, it is known that mucosal barriers such as the airway are continuously exposed to a variety of microorganisms and allergens, and

undergo a continuous process of damage and repair. The release of epithelial cell-derived cytokines, such as IL-33 therefore likely occurs daily. Strict negative regulation of reactivity to IL-33 in antigen-specific helper T cells is therefore beneficial.”

Regarding the second point, this mechanism of negative regulation (high expression of *Dusp10*) could be operating in *Dusp10* high expressing ILC1 and ILC3 cells. This point is also addressed in the revised manuscript (see the 2nd paragraph stated above).

Regarding the third point raised by this reviewer, we discussed the possible function of *DUSP10* in human asthmatic patients with a reference in the revised manuscript (see the first paragraph stated above). In addition, two more paragraphs are included in the revised manuscript (pages 26-28) as follows;

“We identified IL-5-producing ST2^{hi} pathogenic Th2 cells (Tpath2 cells) that likely induce eosinophilic inflammation in nasal polyps of Eosinophilic Chronic Rhinosinusitis (ECRS) patients (Endo Immunity 2015, Nakayama Ann Rev Imm 2017). The antigen-specificity of these cells remains unknown. Similar pathogenic Th2 cell populations with a hPGDS^{hi}CRTH2^{hi} phenotype was identified in the peripheral blood of patients with eosinophilic gastrointestinal diseases and atopic dermatitis (Mitson-Salazar A. et al. J Allergy Clin Immunol. 2016). More recently, Wambre et al. identified allergen-specific human pathogenic Th2 cells with a CRTH2^{hi}CD161^{hi} phenotype that play a role in the pathogenesis of various allergen-specific atopic diseases (Wambre E. et al. Sci Transl Med. 2017). Based on their observations, antigen-induced TCR-mediated activation of the pathogenic Th2 cells appeared to be critical for the pathogenesis of eosinophilic inflammation in these allergic patients. Although further investigation is necessary, DUSP10 may have an important function as a negative regulator of Th2 cytokine production in pathogenic Tpath2 cells in allergic patients.”

“As discussed above, *Dusp10* expression in Th2 cells has been associated with the pathogenesis of allergic disorders including asthma and rhinitis (Seumois et al. 2016). Therefore, small molecules that upregulate DUSP10 or mimic DUSP10 function could be used as new therapeutic agents for the treatment of allergic diseases. Vitamin D receptor (Vdr), a nuclear receptor transcription factor, was shown to control *Dusp10* expression in human adipocyte and prostate cells (Ryynanen et al. *Mol. Nutr. Food. Res.* 2014, and Noon et al. *Cancer Res.* 2006). Moreover, 1,25 (OH)₂D₃, a hormonally active metabolite of Vitamin D, inhibits *in vitro* polarization of primary murine CD4⁺ T cells towards Th2 cell differentiation (Teodora P. et al. *J. Immunol.* 2002). Furthermore, a meta-analysis revealed that the exacerbation of asthma

appeared to be attenuated by the internal use of 1,25 (OH)₂D₃ (Jolliffe DA. et al. *Lancet Respir. Med.* 2017). Thus, although further studies focus on the role of Vdr in the expression of DUSP10 in Tpath2 cells and ILC2s are required, 1,25 (OH)₂D₃ could be involved in the control of *Dusp10* expression, which may lead to the amelioration of the pathology of asthma.”

Reviewer #3 (Airway inflammation, Th2)(Remarks to the Author):

The authors provide a follow up on their previous finding of ST2+ allergen-specific memory-type pathogenic Th2 (Tpath2) cells in allergic eosinophilic airway inflammation. In this study, the authors identify DUSP10, a p38/MAPK/JNK phosphatase, as a key negative regulator of IL-33-induced cytokine production in Tpath2 cells through p38-GATA3 axis to explain the fundamental difference between ILC2 and TH2 cells and why Tpath2 cells do not produce IL-5 and IL-13 in response to IL-33 like ILC2s. This finding is consistent with the current understanding of DUSP10 as a negative regulator of effector T cell (both TH1 and TH2) cytokine expression. As outlined below, the manuscript requires clarification of several issues. General. All flow cytometry analyses include only representative figures and lack. This is not acceptably quantified data including statistical analyses, number of replicates, etc. must be shown.

Response to General:

This point is related to the fifth point raised by the first reviewer. We included quantitative data from more than three independent experiments (**Fig. 1a, 1d, Fig. 3a, 3e, 3h, 4c, and 4f**) in the revised manuscript.

Figure 1. Parts b and c: the fold increase in IL-13 gene and protein expression induced by PMA and Ionomycin (P+I) as compared to IL-7 and IL-33 stimulation in Tpath2 cells is significantly lower than that in flow cytometry analysis in part a, realizing that the quantification method between these techniques is not the same. Please clarify this and include the quantification plot for flow. Part d: The authors have mentioned that ST2hi MPCD4 T cells do not produce IL-5 and IL-13 in response to IL-7 plus IL-33 as ILC2 does, however, the flow analysis also shows that the MPCD4 T cells do not respond much to P+I, the positive control. This makes the claim questionable as the cells do not seem to respond well under any circumstances.

Response:

The difference of fold increase in IL-13 in **Fig. 1a, b, and c** may be due to the duration of stimulation. Prolonged P + I stimulation (in ELISA assay; Fig. 1c) could induce cell death of Tpath2 cells as compared to IL-33 plus IL-7 stimulation. Therefore, fold increase of IL-13 induced by P + I stimulation as compared to IL-33 plus IL-7 stimulation in Fig. 1c may appear to be lower than that in flow cytometry analysis in **Fig. 1a**. We included the quantification of data in Fig. 1a and 1d. Regarding the second point, MPCD4 T cells contain not only Th2 cells but also other ST2⁺ Th subset populations, such as several types of Th1, Th17, and Treg cells. Therefore, the percentage of MPCD4 T cells to respond to P+I stimulation to produce IL-5/IL-13 was lower as compared to that of purified Tpath2 cells. We addressed these issues in the revised manuscript as follows;

“The IL-5 and IL-13 producers in ST2 expressing populations in MPCD4 T cells (ST2^{hi} MPCD4 T cells) are about 10~15%, suggesting that these population may contain non-Th2 ST2^{hi} cells such as Th1, Th17 and Treg cells.”

Figure 3. Part b: in the figure legend, the authors mention total p38 expression, but these data are not shown in the actual figure. In addition, the intensities of the endogenous control, Tub. α , in blots between ILC2 and Tpath2 cells vary significantly, impeding direct comparison of p-p38 expression between the two types of cells when quantification plots are lacking-please perform densitometry to assess the ratio of p-p38 and tubulin alpha to provide more accurate relative quantification. The authors utilize CRISPR-Cas9 system to delete Dusp10 in primary effector cells to study the role of it in IL-33-mediated cytokine production in Tpath2 cells. However, DUSP10 (MKP5) knock-out mice are available and have been used in several studies in the past 1, 2, 3, 4. Is there a reason why CRISPR-Cas9 is preferential to the knock-out mice? It seems that in subsequent experiments, it is more straightforward to using the KO mice, especially in in vivo studies.

Response:

We deleted the description of total p38 in the figure legend. As suggested, we calculated the ratio of p-p38 and tubulin alpha to reflect the relative quantification in the WB analysis more accurately (**Fig. 3b**). Regarding the second point raised by this reviewer, the *Dusp10* KO mice used in studies #1-#4 on the reference list at the end of this letter are whole-KO mice. As shown in reference 1 (Nature 430, 793-797, 2004), *Dusp10* deficiency can affect Th2 cell function during the effector phase. As memory Th2 cells are derived from effector Th2 cells and if any obvious defects in the effector Th2 cells observed, the appropriate interpretation cannot be drawn. Also, a conditional system to delete a certain gene in a specific manner in memory T cells is not available. Therefore, it is

difficult to adequately assess the role of *Dusp10* in the memory-type Tpath2 cell function using whole *Dusp10* KO mice. We therefore used CRISPR/Cas9-induced selective *Dusp10* gene targeting in Tpath2 cells. We believe our CRISPR/Cas9-induced selective *Dusp10* gene targeting system clearly demonstrates the role of *Dusp10* in IL-7/33-mediated function of Tpath2 cells. We also established allergic airway inflammation experimental model systems with cell transfer of CRISPR/Cas9-induced selective *Dusp10* gene targeting, and the role of *Dusp10* in the allergic airway inflammation was addressed.

Figure 4. Part h: the authors claim that overexpression of DUSP10 results in decreased expression of p-p38 in ILC2s compared to the mock-infected ILC2s in response to IL-7 and IL-33 stimulation. However, while the MFI of p-p38 is indeed lower, the shift from the flow histograms between the two experimental groups is very subtle and unconvincing. Again, please provide quantification plots with statistics and also demonstrate the observation through western blots to compliment the flow analysis.

Response:

This point is related to the sixth point raised by the first reviewer. We included quantitative data for **Fig. 3a** (three independent experiments) and **Fig. 3h** (five independent experiments), * $p < 0.05$). Furthermore, we will confirm the expression of p-p38 in *Dusp10*-overexpressing ILC2 by a WB analysis (**Fig. 4g**). We included these results in the revised manuscript.

Figure 5. The authors nicely show the physical interaction among DUSP10, p38 and GATA3 and its dependency on IL-33. It would strengthen the conclusion and add mechanistic insights if the authors can show the role of IL-33 receptor, ST2, in this by using blocking antibody or inhibitor.

Response:

We demonstrated the inhibitory effect of DUSP10 in p38-induced GATA3 activation through the physical association with p38 in Tpath2 cells and ILC2s in the presence or absence of IL-33 stimulation *in vitro*. The association among DUSP10, GATA3, and p38 appeared not to be dependent on IL-33 since IL-33 was not added to the reaction (**Fig. 5a and 5b**). Therefore, we are not able to show the involvement of neutralization of ST2 in the experiment. We believe that DUSP10 suppresses the IL-33-dependent p38 phosphorylation and resulting activation of GATA3. To avoid misunderstanding, we added a sentence in the Result section. "The association among DUSP10, GATA3, and p38 appeared not to be dependent on IL-33 since IL-33 was not added to the reaction."

Figure 6. As mentioned previously, it would further complement the results to use MKP5 KO mice in addition. While the authors examine cellular infiltration to BALF as well as inflammation and mucus production by H&E and PAS staining to evaluate allergic inflammation in both papain and IL-33-induced airway inflammation models, it would strengthen this figure if additional allergic inflammatory parameters are examined as well, such as airway hyperresponsiveness, TH2 cytokine production in the lung and expression of ST2, MUC5AC and GATA3 etc.

Response:

The first point is regarding the usage of *Dusp10* KO mice in the in vivo study. As explained above, the appropriate interpretation can not be drawn on the role of memory Th2 cells. Therefore, we used the CRISPR/Cas9 targeting system in the current study. We established allergic airway inflammation experimental model systems with cell transfer of CRISPR/Cas9-induced selective *Dusp10* gene targeting, and the role of *Dusp10* in the allergic airway inflammation was addressed.

To address the second point, we performed a series of *in vivo* experiments including AHR, ST2/GATA3 expression, and *Muc5ac* expression (qRT-PCR). The degree of AHR in mice that received *Dusp10*-ILC2s was lower than that of mice that received Mock-ILC2s (**Fig. 6i**). Consistent with these observations, *Muc5ac* mRNA expression in the lung tissue was also decreased in mice that received *Dusp10*-ILC2s without affecting the expression levels of *Gata3* and *Il1r1* (**Supplementary Fig. 6f**). We included these results in the revised manuscript.

Overall significance. The findings, if confirmed, are interesting biochemically, but it remains unclear at a practical level what the significance of the findings are. For example, it has now been shown definitively that IL-33 is not important to allergic inflammation and disease in diverse contexts 5, 6.

Moreover, as the authors note, the importance of ILCs in general to human immunity has now been questioned⁷.

Together, these findings seem to challenge the conclusion in the manuscript that, "...DUSP10 may play a critical role in the control of IL-33-associated allergic airway inflammation both [in] mouse and human systems". Ideally, the authors would take on these challenges and attempt to place their findings in this more complete context of the regulation of the type 2 immune response. A more convincing story would further be possible if the authors examined DUSP10 as a potential therapeutic to allergic asthma by examining whether adding DUSP10

or adaptively transferring *DUSP10*-overexpressed ILC2s and TH2 cells rescued mice from developing allergic airway inflammation. Since *GATA3* is incontrovertibly critical for TH2 cell differentiation (and suppressing TH1 cell differentiation), in addition to promoting TH2 cytokine production, the authors could further investigate whether IL-33-*DUSP10*-p38 axis has a role in TH1 and TH2 cell differentiation.

Response:

Regarding the overall significance of the study, we included four new paragraphs in the discussion section in the revised manuscript. This issue is also explained in the response to reviewer #2.

Regarding the importance of IL-33 in allergic inflammation and disease, we do not agree with this reviewer. There is significant evidence that the IL-33/ST2 axis is critical for the induction and pathogenesis of chronic airway inflammation, at least in a certain subset of airway inflammation (Th2 cells in Health and Disease, Nakayama et al. *Ann Rev Imm* 2017). We demonstrated that the IL-33/ST2-p38 MAPK axis is crucial for the induction and enhancement of pathogenicity of memory Th2 cells in allergic airway inflammation in both mice and humans (Endo et al. *Immunity* 2015). We found that IL-5-producing T_{path2} cells express ST2, and exposure to IL-33 induces a significant increase of IL-5 production in memory Th2 cells. IL-33/ST2 signaling upregulates the IL-5 production and ST2 expression in memory Th2 cells through the induction of chromatin remodeling at the *Il5* and *Il1rl1* gene loci without TCR stimulation. *Il33*^{-/-} or *Il1rl1*^{-/-} mice showed decreased eosinophil infiltration into the bronchoalveolar lavage fluid and decreased pulmonary inflammation during airway inflammation. Moreover, recent large-scale meta-genome association studies showed that the *IL33* and *IL1RL1* genes are associated with the onset of asthma (Grotenboer et al. *J. Allergy Clin. Immunol.* 2013). In asthmatic patients, epithelial cells, endothelial cells, and airway smooth muscle cells are thought to be major sources of IL-33 (Molofsky et al. *Immunity* 2015). Polyps from patients with eosinophilic chronic rhinosinusitis (ECRS) patients who frequently are suffering from adult asthma exhibit massive infiltration with CD45RO⁺ memory CD4 T cells that express ST2 and produce large amounts of IL-5 in response to IL-33 (Shinoda et al. *PNAS*, 2016).

In this report, we demonstrate a mechanism (differential expression of *Dusp10*) that differentiates the innate function such as cytokine-induced cytokine expression characteristic to ILC2s from antigen-dependent adaptive Th2 cell function. As we stated in the discussion section in the revised manuscript, we

would like to extend this finding to the research in the development of the agents for intractable eosinophilic inflammation such as severe asthma.

References

1. Zhang, Y. et al. *Regulation of innate and adaptive immune responses by MAP kinase phosphatase 5. Nature* 430, 793-797 (2004).
2. Qian, F. et al. *Map kinase phosphatase 5 protects against sepsis-induced acute lung injury. American journal of physiology. Lung cellular and molecular physiology* 302, L866-874 (2012).
3. Cheng, Q. et al. *MAPK phosphatase 5 deficiency contributes to protection against blood-stage Plasmodium yoelii 17XL infection in mice. Journal of immunology* 192, 3686-3696 (2014).
4. Shi, H. et al. *Improved regenerative myogenesis and muscular dystrophy in mice lacking Mkp5. The Journal of clinical investigation* 123, 2064-2077 (2013).
5. Hoshino, K. et al. *The absence of interleukin 1 receptor-related T1/ST2 does not affect T helper cell type 2 development and its effector function. J Exp Med* 190, 1541-1548 (1999).
6. Vannella, K.M. et al. *Combinatorial targeting of TSLP, IL-25, and IL-33 in type 2 cytokine-driven inflammation and fibrosis. Sci. Transl. Med.* 8, 337ra365 (2016).
7. Vely, F. et al. *Evidence of innate lymphoid cell redundancy in humans. Nat Immunol* 17, 1291-1299 (2016).

REVIEWERS' COMMENTS:

Reviewer #1 (Remarks to the Author):

The authors performed an extensive revision of the manuscript, included additional data and clarified the description of the experimental conditions. Overall the revision satisfactorily addressed my different concerns

Reviewer #2 (Remarks to the Author):

I am content with the response of the authors.